# Protein corona formed on lipid nanoparticles compromises delivery efficiency of mRNA cargo

Elizabeth Voke [1], Mariah L. Arral[2], Henry J. Squire [1], Teng-Jui Lin [1], Lining Zheng[1], Roxana Coreas[1], Alison Lui[1], Anthony T. Iavarone[3], Rebecca L. Pinals [4,5,6,7] ✉, Kathryn A. Whitehead [2,8] ✉ & Markita P. Landry [1,3,9] ✉

Lipid nanoparticles (LNPs) are the most clinically advanced nonviral RNA-delivery vehicles, though challenges remain in fully understanding how LNPs interact with biological systems. In vivo, proteins form an associated corona on LNPs that redefines their physicochemical properties and influences delivery outcomes. Despite its importance, the LNP protein corona is challenging to study owing to the technical difficulty of selectively recovering soft nanoparticles from biological samples. Herein, we develop a quantitative, label-free mass spectrometry-based proteomics approach to characterize the protein corona on LNPs. Critically, this protein corona isolation workflow avoids artifacts introduced by the presence of endogenous nanoparticles in human biofluids. We apply continuous density gradient ultracentrifugation for protein-LNP complex isolation, with mass spectrometry for protein identification normalized to protein composition in the biofluid alone. With this approach, we quantify proteins consistently enriched in the LNP corona including vitronectin, C-reactive protein, and alpha-2-macroglobulin. We explore the impact of these corona proteins on cell uptake and mRNA expression in HepG2 human liver cells, and find that, surprisingly, increased levels of cell uptake do not correlate with increased mRNA expression in part due to protein corona-induced lysosomal trafficking of LNPs. Our results underscore the need to consider the protein corona in the design of LNP-based therapeutics.

Lipid nanoparticles (LNPs) are advanced nonviral ribonucleic acid (RNA) delivery vehicles for clinical applications. These LNPs function to protect RNA against degradation during transit into cells and facilitate endosomal escape for the delivery of their RNA cargo following cell internalization[1–5]. The clinical success of these therapeutics has been demonstrated by Alnylam Pharmaceuticals' LNPs loaded with small interfering RNA (siRNA) to treat liver amyloidosis[6] and messenger RNA (mRNA)-based vaccines against SARS-CoV-2 from Moderna and Pfizer/BioNTech[7]. Emerging applications of mRNA delivery additionally include protein replacement therapy, immunotherapy, and

[1]Department of Chemical and Biomolecular Engineering, University of California, Berkeley, Berkeley, CA, USA. [2]Department of Chemical Engineering, Carnegie Mellon University, Pittsburgh, PA, USA. [3]California Institute for Quantitative Biosciences (QB3), University of California, Berkeley, Berkeley, CA, USA. [4]Picower Institute for Learning and Memory, Massachusetts Institute of Technology, Cambridge, MA, USA. [5]Department of Brain and Cognitive Sciences, Massachusetts Institute of Technology, Cambridge, MA, USA. [6]Department of Chemical Engineering, Stanford University, Stanford, CA, USA. [7]Sarafan ChEM-H Institute, Stanford University, Stanford, CA, USA. [8]Department of Biomedical Engineering, Carnegie Mellon University, Pittsburgh, PA, USA. [9]Department of Department of Neuroscience, University of California, Berkeley, Berkeley, CA, USA. ✉e-mail: rpinals@stanford.edu; kawhite@cmu.edu; landry@berkeley.edu

gene editing[3,4,8]. Despite the success of locally administered vaccines, achieving organ- and cell-type specific LNP delivery outside the liver from intravenous administration remains challenging. Given the therapeutic promise of LNPs, the development of new formulations with enhanced potency and targeted delivery outcomes has been a key area of focus for furthering clinical translation, garnering significant commercial interest[9].

To improve LNP potency and develop formulations for selective organ- or cell-type targets, large LNP formulation libraries with subsequent in vivo screens are conventionally implemented for accelerated materials discovery[1,10,11]. The primary focus of the field has been engineering LNPs through formulation alterations including changes in lipid structure[12–14], the introduction of targeting ligands such as antibodies to the surface[15], and tuning polyethylene glycol (PEG) density[16]. While this work has shown success in developing more potent delivery vehicles[13,17] and delivery to extrahepatic tissues[18–21], the mechanisms behind the increased potency from formulation changes or how modification to LNP composition alters organ tropism remain unclear. This lack of mechanistic understanding limits future rational design. Moreover, these screens fail to predict how changes in particle function in the context of in vitro screens will translate to in vivo LNP efficacy[22,23].

Evidence has established a potential relationship between protein recruitment to the LNP surface and organ targeting[14,24–26] and functionality[27,28], necessitating further characterization of the interactions between proteins and LNPs. As such, we seek to explore how the LNP identity is redefined by the spontaneous adsorption of biofluid proteins, and how these LNP corona proteins impact their function. Upon injection, nanoparticles encounter various biological tissues and compartments. Biomolecules such as proteins spontaneously interact with the nanoparticles and form an associated protein corona[29–32]. Proteins with a strong affinity for the particle surface form a "hard corona," while more loosely associated proteins form a dynamic "soft corona"[30]. These corona proteins modify nanoparticle function and localization in vivo, as this outer protein layer changes how nanoparticles interact with cell-surface receptors, impacting biodistribution[33,34] and cell-specific uptake[35,36]. Upon systemic injection, most nanoparticles are cleared by the liver and, in particular for LNPs, adsorption of apolipoprotein E (ApoE) facilitates interactions with low-density lipoprotein receptors on the surface of hepatocytes to mediate intracellular delivery[25]. We hypothesize that protein corona formation impacts the core functions of LNPs: delivery localization, cell internalization, and endosomal escape, all of which are required for mRNA cargo delivery.

In this work, we applied a quantitative, label-free mass spectrometry-based proteomics workflow that leverages continuous density gradients to probe the nano-bio interface of LNPs in human blood plasma. Our approach accounts for the presence of native particles in the proteomic analysis of the corona, without modification of the LNP formulation or surface. We provide clarity on best practices

for sample preparation to reproducibly collect highly enriched LNP corona proteins, and through this approach, consistently find proteins associated with lipid transport and metabolism enriched in the corona. We identify a small set of proteins that form the putative hard corona on LNPs and examine how they influence LNP transfection. By studying LNPs with pre-formed protein coronas, we discovered a mismatch between internalization and mRNA expression: certain corona proteins increased cellular uptake of LNPs by five-fold but had no effect on mRNA expression. This work establishes a framework to reliably characterize proteins enriched on the LNP surface and shows that a subset of these proteins (e.g., vitronectin) significantly affect LNP uptake into cells and compromise LNP transfection efficiency. Connecting protein corona formation on LNPs to cellular delivery outcomes provides insight into the biomolecular mechanisms that limit LNP transfection, particularly the low endosomal escape efficiency estimated at 2%[2]. Our findings suggest that increased cellular uptake does not necessarily improve transfection, especially when the protein corona may hinder endosomal escape. We propose that the protein corona influences both LNP uptake and intracellular trafficking. However, pinpointing the specific proteins that are strongly and consistently enriched in the LNP corona remains experimentally challenging, limiting our ability to assess their influence on key steps of cargo delivery.

## Results

### Limitations of current methods for protein corona characterization on LNPs

The development of methods to study protein-LNP interactions is difficult due to the similar properties of lipid-based nanomaterials and the nanoparticles intrinsically present in the biological fluids they will encounter in vivo, such as plasma in the context of intravenous administration. Broadly, biological fluids are mixtures of many constituents including individual biomolecules and biological particles, with diameters on the scale of nanometers to micrometers. Plasma, for example, contains proteins such as serum albumin, the most abundant protein in plasma, and endogenous particles including extracellular vesicles and lipoproteins. Such particles are primarily composed of lipids and proteins, and have diameters ranging from 7 to 1200 nm[37,38]. LNPs often have diameters ranging from 30 to 200 nm[39], and protein corona formation would likely increase LNP hydrodynamic size[40]. Effective isolation of protein-LNP complexes from biological fluids thus requires separation from these endogenous particles while also maintaining stable LNPs with an intact corona[39]. However, selective LNP isolation has remained a major challenge because these native particles have similar sizes and compositions relative to protein-LNP complexes (Fig. 1a)[39,41,42]. Additionally, attempts to isolate protein-LNP complexes may impact particle stability and corona integrity[39,43].

A further challenge of isolating protein-LNPs is the low density of these soft nanoparticles. For denser substrates such as polymeric nanoparticles, standard centrifugation is sufficient to pellet protein-

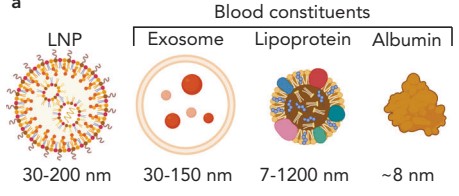
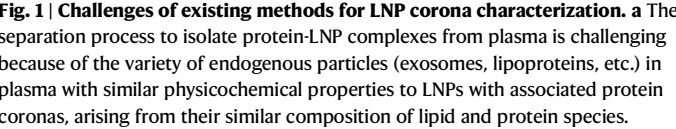
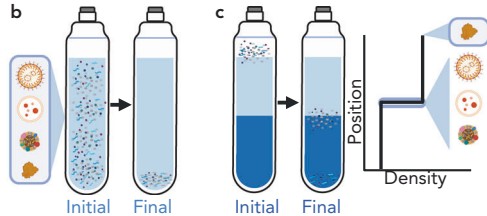

**Fig. 1 | Challenges of existing methods for LNP corona characterization. a** The separation process to isolate protein-LNP complexes from plasma is challenging because of the variety of endogenous particles (exosomes, lipoproteins, etc.) in plasma with similar physicochemical properties to LNPs with associated protein coronas, arising from their similar composition of lipid and protein species. Illustrations demonstrating **b** why ultracentrifugation (that pellets all particles) and **c** discrete sucrose gradients (that isolate LNPs at the interface of two different density solutions) fail to effectively separate LNPs from biofluid-derived particles. Created in BioRender. Voke, E. (2025) https://BioRender.com/58u4s4t.

nanoparticle complexes from free proteins that remain suspended in-solution, leading to well-established protein corona-isolation techniques[44]. In contrast, the low density of LNPs renders these particles buoyant in common buffers such as phosphate-buffered saline (PBS), even upon incubation with biological fluids. This buoyancy prevents LNP pelleting via tabletop centrifugation. We demonstrate this challenge by characterizing a potent LNP synthesized with the lipidoid, $306O_{10}$, as a model LNP[12,19]. We used dynamic light scattering (DLS) to measure the hydrodynamic diameters of constituents in the supernatant post centrifugation for 30 minutes at 4 °C and $20,000 \times g$ (Supplementary Fig. 1a). Alternatively, higher g-forces (ultracentrifugation) have been shown to result in aggregation[45] or disruption[46] of these low-density lipid-based particles, as we also confirm by ultracentrifugation for 2 hours at 4 °C and $160,000 \times g$ (Supplementary Fig. 1a). Ultracentrifugation also fails to provide LNP separation from biofluid-derived particles, as all particles eventually sediment to the bottom of the tube at longer time scales (Fig. 1b). Other techniques such as size exclusion chromatography (SEC) generally preserve particle stability but fail to effectively separate endogenous particles[45]. Additionally, sucrose cushions, which isolate LNPs at an interface between fluids of different densities, trap endogenous particles with the protein-LNP complex (Fig. 1c) and a lack of plasma controls makes it challenging to distinguish between proteins interacting with LNPs and proteins interacting with endogenous similarly-sized particles, such as exosomes.

Some methods have been developed in recent years such as photoaffinity-based[47], antibody-based[48], and magnetic-based[49] isolations that are high-throughput and include wash steps to remove native particles. However, photoaffinity-based and magnetic-based approaches require modifications of the lipid-based formulations that may impact the corona proteins identified, whereas antibody-based pulldowns targeting PEG may be biased by PEG desorption from the LNP surface[50]. These methods have been highly valuable in enabling larger formulation screens, whereas a method that does not alter LNP-corona formation or rely on PEG presence is still needed for further mechanistic studies of protein-LNP complexes. Another approach that can be used to study LNP corona proteins while avoiding the contribution of endogenous particles is to use plasma depleted of lipoproteins, yet the use of depleted plasma fails to capture interactions between apolipoproteins and the LNP, which are often associated with the mechanism of LNP uptake, such as ApoE[25].

Density gradient ultracentrifugation (DGC) is a promising method that is gentle on the protein corona, does not require changes to the LNP formulation, and enables relative separation from more dense lipoproteins. Within a density gradient, the medium may vary in density in a linear or stepwise manner depending on the medium selected and the centrifugation conditions. As samples are centrifuged in a density gradient, lower-density particles, including LNPs, float towards the top, while denser plasma protein components, like serum albumin, sink to the bottom. Previous studies characterizing the LNP corona using this approach separate particles at relatively short time scales (3–4 hours)[28,51] and thus fail to effectively separate protein-LNP complexes from the more abundant plasma proteins and endogenous nanoparticles (Supplementary Fig. 1b). As a result, protein corona characterization from these studies includes proteins recovered both from LNPs and from biofluid-derived particles, making it difficult to assess which of these proteins originated from the LNP corona itself[18,28,51]. In contrast, most methods for separating exosomes from biofluids within a density gradient use longer centrifugation times of ~16–24 hours to accomplish a clean separation[52,53]. Here, we postulated that by (1) providing adequate separation time to isolate protein-LNP complexes and (2) accounting and correcting for native particle contamination, we could identify and quantify the presence of proteins that adsorb to the LNP surface in human biofluids.

## Improved workflow for protein corona isolation from LNPs

To address the limitations of current techniques, we developed a workflow that employs a continuous linear density gradient to isolate protein-LNP complexes, followed by proteomic analysis (Fig. 2). In this workflow, we incubated LNPs with pooled human blood plasma for 1 hour at 37 °C before loading onto the bottom of a six-layer iodixanol gradient and centrifuging for 16 hours at 36 kilorotations per minute (krpm) (Fig. 2a). This workflow was inspired by methods used in the exosome field to separate subpopulations of exosomes[54,55]. Unlike discrete gradients with step-change differences in density (Fig. 1c), an iodixanol gradient linearizes over the course of centrifugation[56], forming a continuous gradient that enables a finer degree of separation in fractions throughout the linear region of the tube. We confirmed the stability of the LNPs after density gradient centrifugation with DLS, which showed colloidally stable particles (Supplementary Fig. 2). As an additional quality-control, we checked the density throughout the gradient via refractive index and absorbance to ensure that density varies linearly through the tube (Supplementary Fig. 3,4).

After centrifugation, we used fluorescence measurements to track LNP localization and selected fractions for collection. Based on our DLS measurements (Supplementary Fig. 1a) and prior DLS characterization of LNPs[17], we determined that our synthesized LNPs possessed a low polydispersity and a narrow diameter range. This suggested that LNPs would distribute within a small range of fractions within the iodixanol gradient. We identified fractions containing LNPs by synthesizing an LNP sample with a fluorescently tagged lipid (1,2-dioleoyl-sn-glycero-3-phosphoethanolamine-N-(lissamine rhodamine B sulfonyl)) and running the fluorescent LNPs through the iodixanol gradient (Fig. 2b). 0.5-mL fractions were collected top to bottom and fluorescence was measured to quantify LNP localization (Fig. 2c) as well as absorbance to confirm linearity of the iodixanol gradient (Supplementary Fig. 4). Based on our fluorescence measurements, we found that ~68% of LNPs localized within fractions 2−6 of the iodixanol gradient, denoted as a single sharp peak in the early gradient fractions (Fig. 2c, Supplementary Table 1). We observed a broad minor second fluorescence peak at higher fraction numbers (Fig. 2c), which is likely due to fluorophores dissociating from the LNP as previously demonstrated[57]. The autofluorescence of proteins in blood plasma was found to be negligible.

To examine the degree of separation from lipoproteins, which are endogenous particles that confound LNP protein corona results, we quantified the presence of total cholesterol as a key lipoprotein constituent throughout the gradient (Fig. 2d). We found that most cholesterol is present in fractions 5–10 and later fractions, which have limited overlap with the localization of the LNPs. To further validate the localization of lipoproteins, we quantified the presence of the most abundant apolipoprotein in human plasma, apolipoprotein AI (ApoA-I). We confirmed that 99.5% of ApoA-I localizes to fractions 12–24 through an enzyme-linked immunosorbent assay (ELISA; Supplementary Fig. 5). Previous work has already determined that exosomes, another type of endogenous particle, are not present in the first 5 fractions of the density gradient[54]. By pooling fractions 2–6 for characterization via liquid chromatography-tandem mass spectrometry (LC-MS/MS), the concentration of LNPs was maximized relative to amounts of native particles present in the control sample. We elected to keep the marginal fractional overlap between localization of the LNPs and lipoproteins in fraction 5–6 to have sufficient protein amounts for proteomic processing and to avoid biasing the recovery of proteins from LNPs of slightly smaller size or higher density. Importantly, our control sample accounts for the fractional overlap of LNPs and lipoproteins through proteomic comparison. This process of fraction selection allows us to minimize contributions of endogenous blood particles and predominantly focus on LNP corona proteins for downstream analysis.

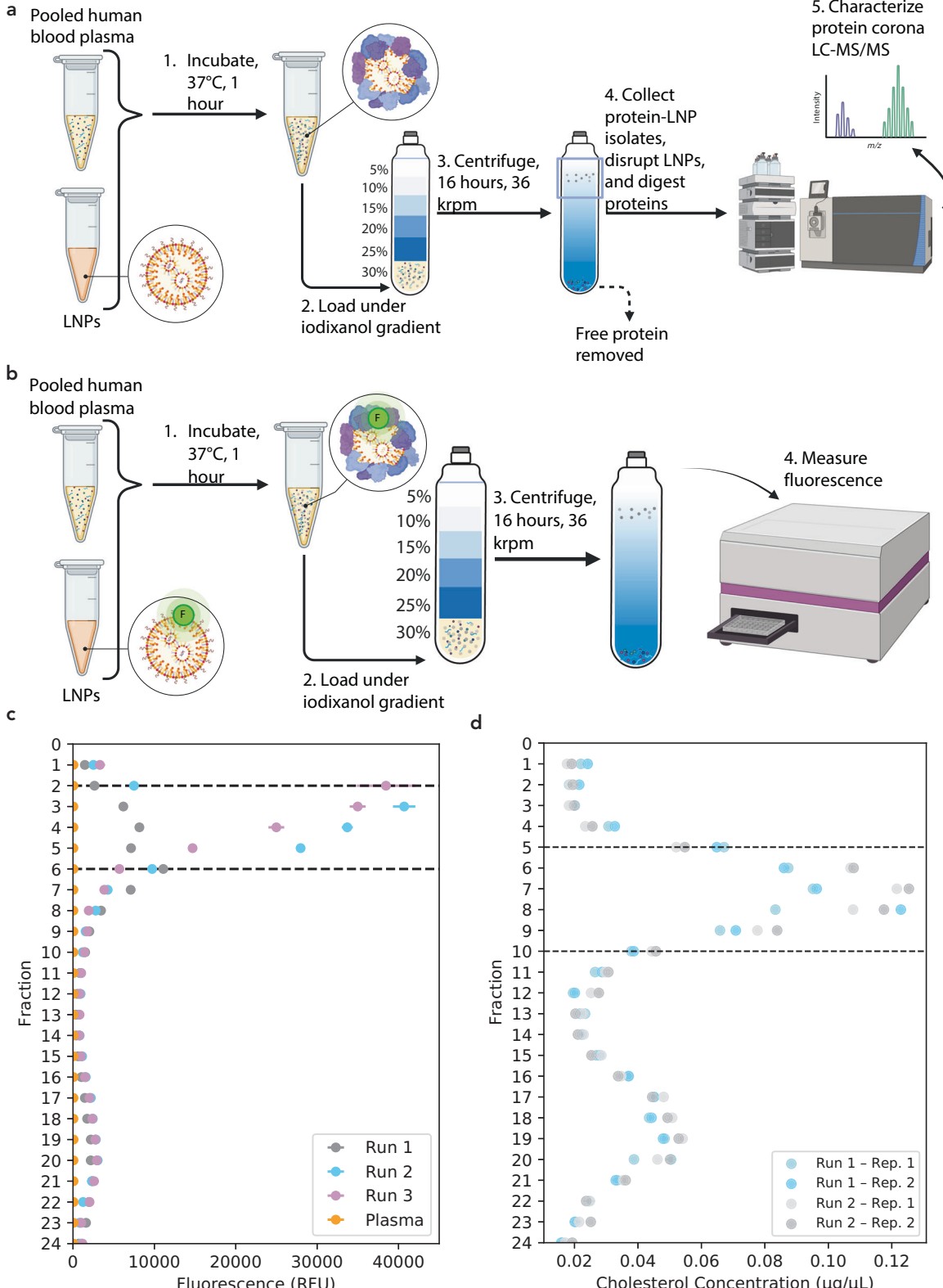

## Proteomic characterization of the protein corona isolated from LNPs

To selectively characterize the LNP corona, we account for the presence of native biological particles through fraction selection and normalization. We normalize by similarly separating a plasma-alone sample with DGC and submitting the same selected fractions as those with LNPs present for proteomic characterization (Fig. 3a). Through this analysis, we identified 56 proteins in the LNP protein corona and in the plasma-alone sample, which then allowed us to calculate the protein abundance fold change of protein-LNP samples relative to plasma control fractions. Peptide coefficients of variation (CV)% distribution for LNP and plasma samples (Fig. 3b) show low variation with a median CV% of 11.8 and 19.0 for the LNP and plasma samples, respectively. Out of the 56 identified proteins, 53 proteins were found to have significant

**Fig. 2 | Proteomics workflow for label-free, quantitative protein corona profiling on LNPs. a** LNPs were incubated with pooled human blood plasma for 1 hour at 37 °C then mixed with the low osmolarity density gradient medium, iodixanol, to a final concentration of 30% iodixanol before being loaded under five distinct layers of iodixanol (25%, 20%, 15%, 10%, and 5%) and centrifuged for 16 hours at 36,000 rpm. 0.5-mL fractions were collected from the top to the bottom and selected fractions were processed for LC-MS/MS characterization. Created in BioRender. Voke, E. (2025) https://BioRender.com/58u4s4t. **b** LNPs were tagged with lissamine rhodamine, incubated with blood plasma, and loaded under an iodixanol gradient with the same isolation workflow conditions. Created in

BioRender. Voke, E. (2025) https://BioRender.com/58u4s4t. **c** Fluorescence measurements of fluorescently tagged LNPs after the DGC isolation workflow reveal that 0.5-mL fractions 2–6 (dotted lines) in the density gradient have the maximum number of LNPs. Excitation/emission wavelengths of 560/580 nm were used to detect lissamine rhodamine-tagged LNPs. $N = 4$ technical replicates, $n = 3$ biological replicates. Data are presented as mean values with standard deviation. **d** Average total cholesterol quantification of plasma-alone gradient fractions collected after DGC isolation workflow show that lipoproteins within plasma are present primarily among fractions 5–10 (dotted lines). $N = 2$ technical replicates, $n = 2$ biological replicates.

differences (false discovery rate (FDR) corrected *p* value (*q* value) <0.05) between the protein-LNP sample and the plasma control sample, with 39 proteins enriched in the LNP corona and 14 proteins depleted (Fig. 3c). The enriched subset of proteins is relatively small compared to existing literature on protein-LNP complexes, suggesting that our approach removes proteins that are abundant in plasma alone but not necessarily relevant to the protein corona. Other label-based corona-isolation methods have identified a similar number of proteins enriched in the LNP protein corona[47,49]. We also attempted density gradient centrifugation using previously reported centrifugation conditions (4 hours), which yielded high levels of serum albumin in the fractions where the LNPs localized (Supplementary Table 2). As such, our method of using a longer centrifugation time at a higher speed with a more robust density gradient layering technique reduces the presence of serum albumin in the fractions of interest, suggesting a more effective separation of protein-LNP complexes from free plasma proteins with longer centrifugation times.

We next categorized proteins enriched in the LNP corona based on their gene ontology, specifically, their biological process, cellular component, and molecular function (Fig. 3d). We found that the biological processes of these proteins are associated with both the innate and adaptive immune responses, as well as lipid transport and metabolism. As anticipated, the cellular component characterization of these proteins reveals their associations with the extracellular space, exosomes, and microparticles. Their molecular functions were associated with lipid-binding, immunoglobulin receptor-binding, and heparin-binding functions. Further analysis also revealed that enriched LNP-corona proteins were involved in biological pathways including cholesterol metabolism (Fig. 3e) and components in apolipoproteins (Fig. 3f). Despite ApoA-I and apolipoprotein A-II (ApoA-II) being the two most abundant apolipoproteins in blood plasma[58], we do not identify ApoA-I or ApoA-II as enriched in the protein corona, suggesting we are selectively identifying apolipoproteins that interact with LNPs. Additionally, we find that proteins implicated in complement and coagulation cascades are enriched in the corona phase (Fig. 3e).

We compared the fold change in protein abundance relative to plasma alone (Fig. 3g), which revealed enriched proteins such as c-reactive protein (CR) that have a high affinity for the LNPs. In previous methods, this low-abundance protein would be challenging to identify as an enriched protein due to high levels of contamination from high-abundance proteins, such as serum albumin and apolipoproteins. We also found that vitronectin, a cell adhesion and spreading factor that interacts with glycosaminoglycans and proteoglycans, is highly enriched in the LNP protein corona, in agreement with prior work[28].

To highlight the merits of this approach, we examined the relationship between proteomic analyses that considered the relative protein abundance only upon LNP incubation and our approach that quantifies differences between the LNP sample and a biofluid control (Supplementary Fig. 6a). In previous LNP corona work[28], the relative abundance was reported as the percent abundance of each protein identified in the LNP experiment without a biofluid-alone control. In contrast, we quantify the absolute protein abundance and report the

fold change in the LNP sample relative to the biofluid control. We found a near-zero and negative correlation for our data analysis (fold change relative to plasma) and previous approaches for reporting top-enriched proteins (relative abundance (%)) for all identified proteins and apolipoproteins, respectively (Supplementary Fig. 6b, c). These results suggest that examining the relative abundance in an LNP sample is not sufficient for selective identification of proteins that comprise the LNP protein corona. Analyzing the LNP sample by highest relative abundance (%) likely biases toward higher abundance plasma proteins. As such, proteins that are more abundant in the biofluid, including ApoA-I, may appear highly enriched in the corona. Therefore, characterizing the LNP protein corona with centrifugation-based approaches by only considering the most abundant proteins in the corona is less accurate and is largely overwhelmed by proteins introduced by particles native to plasma, and not by interaction with the LNPs.

### Effect of proteins enriched in the LNP corona on LNP function
Ultimately, we are interested in studying how proteins consistently enriched in the LNP protein corona affect LNP transfection efficiency. Our analysis thus far highlights the proteins most enriched in the LNP protein corona from three technical replicates. Our group has previously shown that experimental replicates, particularly those performed on different days and analyzed at different LC-MS/MS core facilities, exhibit very high variability, with <2% common proteins identified from different LC-MS/MS core facilities from otherwise identical protein corona samples[59,60]. Therefore, we performed three independent experimental replicates of our protein isolation workflow to assess the true variation within our method. To do so, we compared enriched proteins from samples processed in parallel, which have limited LC-MS/MS instrument variation, and samples processed via LC-MS/MS independently across different weeks, each with three replicates of the isolation workflow. This experiment ensures that proteins we find across several independent and time-separated replicate datasets are consistently enriched in the corona. Samples processed in parallel (Supplementary Fig. 7) show similar proteins enriched in the corona. We therefore conclude that these proteins have a high association with the LNP surface, and their consistent enrichment through the density gradient isolation strategy suggests that these proteins are likely "hard corona" proteins.

We next analyzed specifically which subset of proteins is consistently enriched in the LNP corona across the different batches processed by LC-MS/MS (Supplementary Table 3). This analysis reduces the variability contributed by the LC-MS/MS method itself in detecting low-abundance corona proteins and enables us to study consistently enriched proteins in greater mechanistic depth: alpha-2-macroglobulin, C-reactive protein, and vitronectin, as summarized in Table 1. Thus, by including independent batches of experimental runs that include both technical and experimental replicates, analyzing our data relative to the plasma control, and using a continuous iodixanol gradient protocol, we reproducibly measure and identify proteins that are consistently enriched in the LNP protein corona. Having identified several LNP-corona proteins

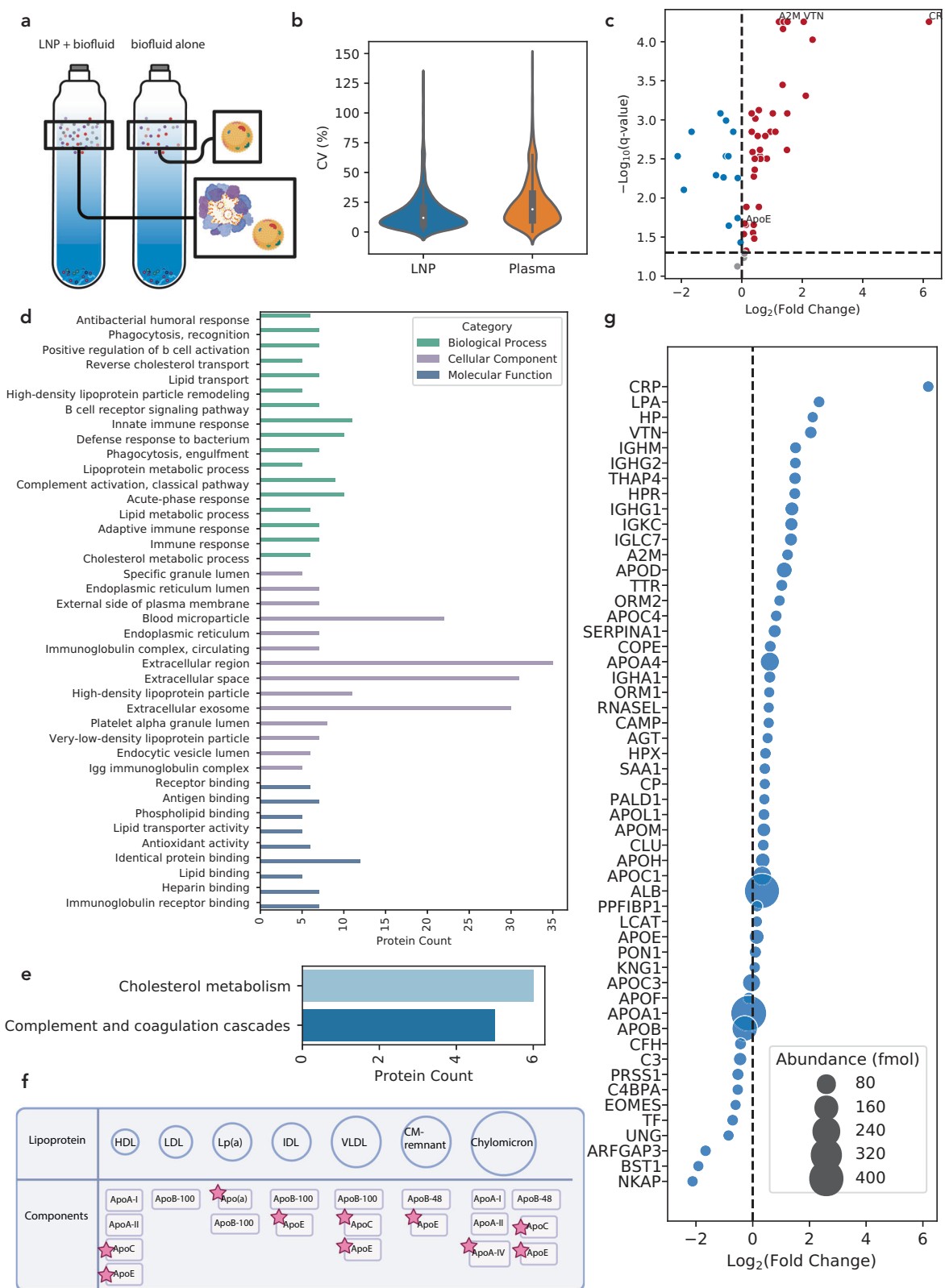

consistently observed with high enrichment in the protein corona across independent batches and parallel replicates, as summarized in Table 1, we sought to study their effects on LNP cellular interactions and function. Additionally, we included ApoE in our downstream studies because of its putative relevance to LNP cellular internalization, despite the variability with which we measured its presence in the corona (Supplementary Fig. 7).

## Protein-nanoparticle interactions impact nanoparticle functionality

To understand the effect of corona proteins on LNP-mediated mRNA delivery, we assessed the mRNA delivery and protein expression efficiency in cell culture for LNPs with coronas pre-formed using proteins identified in our proteomic analysis (Table 1). We considered both single protein-LNP coronas and an LNP corona formed from the

**Fig. 3 | Proteomic analysis of the LNP protein corona. a** Normalization across density gradient fractions enables proteomic analysis that accounts for native lipoproteins found in plasma. Created in BioRender. Voke, E. (2025) https://BioRender.com/58u4s4t. **b** Violin plot of peptide coefficient of variation (CV) analysis shows low variation in peptide quantification during LC-MS/MS ($n = 3$ technical replicates). The overlaid box-and-whisker plot is defined as follows: the white dot indicates the median; the box spans the interquartile range (IQR), from the 25th to 75th percentiles; and the whiskers extend from the IQR to the minimum and maximum values within 1.5×IQR. **c** Log2 fold change of LNP-corona proteins discovered via LC-MS/MS vs. negative log10 of the $q$ value, showing nonsignificant proteins in gray, significantly enriched corona proteins in red, and significantly depleted corona proteins in blue ($n = 3$ technical replicates). **d** Gene Ontology analysis of enriched corona proteins and **e** KEGG pathway analysis of enriched corona proteins are shown for $p$ values < 0.05. **f** Enriched proteins mapped to lipoprotein components with identified proteins starred. Created in BioRender. Voke, E. (2025) https://BioRender.com/58u4s4t. **g** Log2 fold change of LNP-corona proteins, with bubble size denoting femtomolar (fmol) abundance. Please see the Methods section for full details on analysis and significance.

## Table 1 | Proteins enriched in the LNP corona chosen for in vitro study

| Protein | Entry | Function | Ref. |
|---|---|---|---|
| Alpha-2-macroglobulin | A2M | Inhibits all four classes of proteinases | 88 |
| Apolipoprotein E | ApoE | Facilitates interactions with low-density lipoprotein receptors for lipid transport | 25 |
| C-reactive protein | CR | Activates the complement pathway | 65 |
| Vitronectin | VTN | Cell adhesion and spreading factor | 66 |

combination of the selected top-enriched proteins. LNPs were loaded with a luciferase mRNA that provides a quantitative, luminescent readout upon successful luciferase mRNA translation to protein. 2 μg of each protein (0.05 ng mRNA: 1 ng protein, 0.01 mg/mL protein), an amount that is in excess of its presence in the corona as measured by LC-MS/MS (Supplementary Table 4), was incubated for 1 hour at 37 °C with each LNP formulation before LNP introduction to HepG2 human liver cells in serum-free media for attempted transfection. Of note, protein concentrations are within the same order of magnitude as native plasma protein concentrations, except for A2M, which is more highly abundant in plasma (Supplementary Table 5). We measured the increase in the hydrodynamic radii of the LNPs after the protein incubations, confirming that these select proteins form an associated LNP corona (Supplementary Table 6) prior to their introduction to cells.

The output luminescence was measured with a plate reader after 24 hours and compared across protein corona conditions (Fig. 4a). We found that LNP protein coronas formed from proteins ApoE, A2M, and the protein mixture of all four proteins together did not have a significant impact on mRNA expression levels relative to LNPs without a pre-formed protein corona. In contrast, LNPs with VTN or CR pre-formed coronas showed decreased mRNA expression relative to LNPs without a pre-formed corona (Fig. 4b). We observed an approximately 50% decrease in mRNA expression for LNPs with a VTN corona and ~90% decrease in mRNA expression for LNPs with a CR corona. We also confirmed that LNP uptake occurs through the anticipated pathway of endocytosis. To test this, we measured mRNA expression in HepG2 human liver cells treated with Dynasore and LNPs that were either bare or protein corona-coated. Dynasore functions as an inhibitor of clathrin-coated pit-mediated endocytosis as well as fast endophilin-mediated endocytosis, a dynamin-dependent, clathrin-independent pathway for rapid ligand-driven endocytosis[61]. We found that mRNA expression from LNPs both with and without the pre-incubated corona was entirely reliant on dynamin-dependent endocytosis pathways (Supplementary Fig. 8a). These effects were observed at inhibitor concentrations that did not influence cell viability (Supplementary Fig. 8b).

We next investigated the concentration dependence of the VTN corona on mRNA expression. A dose-response experiment reveals that pre-incubating LNPs with VTN protein concentrations above 0.005 mg/mL (1000 ng added) exhibited significantly decreased mRNA expression efficiency relative to LNPs without a protein corona (Fig. 4c). This protein concentration at 0.005 mg/mL represents a lower VTN concentration than found in native plasma (Supplementary Table 5). We also considered that LNPs with pre-formed coronas may affect cell viability and thus indirectly affect transfection efficiency. However, we found that the pre-formed single-constituent protein coronas had no significant impact on cell viability (Fig. 4d), demonstrating that the decrease in mRNA expression is not due to lower cell viability.

From these cell transfection expression experiments, we conclude that protein-LNP interactions impact the ability of LNPs to deliver mRNA into cells' cytoplasm for transfection. We hypothesize that pre-formed coronas on LNPs, which compromise LNP transfection efficiency, may show altered interactions with cells during cargo delivery. To investigate how pre-formed LNP coronas affect LNP-cell interactions, we first considered how pre-formed LNP coronas influence LNP cellular uptake, an essential step for mRNA expression. We used confocal microscopy to visualize and quantify differences in cell uptake of LNPs loaded with Cy5-tagged mRNA, each with a pre-formed, single-constituent protein corona formed with ApoE, VTN, A2M, CR, or a mixture of all corona proteins (Fig. 5a, b). We specifically selected the mRNA for fluorophore-based visualization to enable tracking of the functional cargo, because fluorescent tagging of other LNP constituents, such as lipids, may result in exchange with the surrounding environment[57]. We quantified the Cy5 signal found within the cell membrane mask as a proxy for LNP uptake within the cells and normalized this signal per cell by the nuclei count. We found significantly increased Cy5 signal per cell for LNPs with pre-formed ApoE, VTN, or mixed protein coronas. We observed no significant difference in Cy5 signal per cell for LNPs with pre-formed A2M or CR coronas (Fig. 5c). No signal was observed from protein-only controls added to cells (Supplementary Fig. 9a). In the case of the ApoE-LNP corona, we found five-fold higher levels of Cy5 signal per cell compared to the LNPs without a pre-formed protein corona. This increase in uptake of LNPs with a pre-formed ApoE corona is supported by previous literature that associates ApoE with more internalization in hepatocytes via receptor-mediated uptake[25]. Additionally, LNPs with pre-formed VTN coronas had four-fold observed higher Cy5 signal per cell than cells treated with LNPs alone. However, unlike ApoE, VTN is not associated with increased uptake in HepG2 cells. VTN is a cell adhesion protein that may drive LNP adhesion to the outer cell surface. Interestingly, we also observe a fourfold higher Cy5 signal per cell for the mixture of corona proteins.

We investigated whether the increase in Cy5 signal per cell may be due to LNPs associating with the outer membrane of the cell rather than being fully internalized. Images were collected from adherent cells with a 4.5 μm offset from the bottom of the cell, enabling visualization through an intermediate slice of each cell (Supplementary Fig. 10). This approach enables us to observe LNP association within

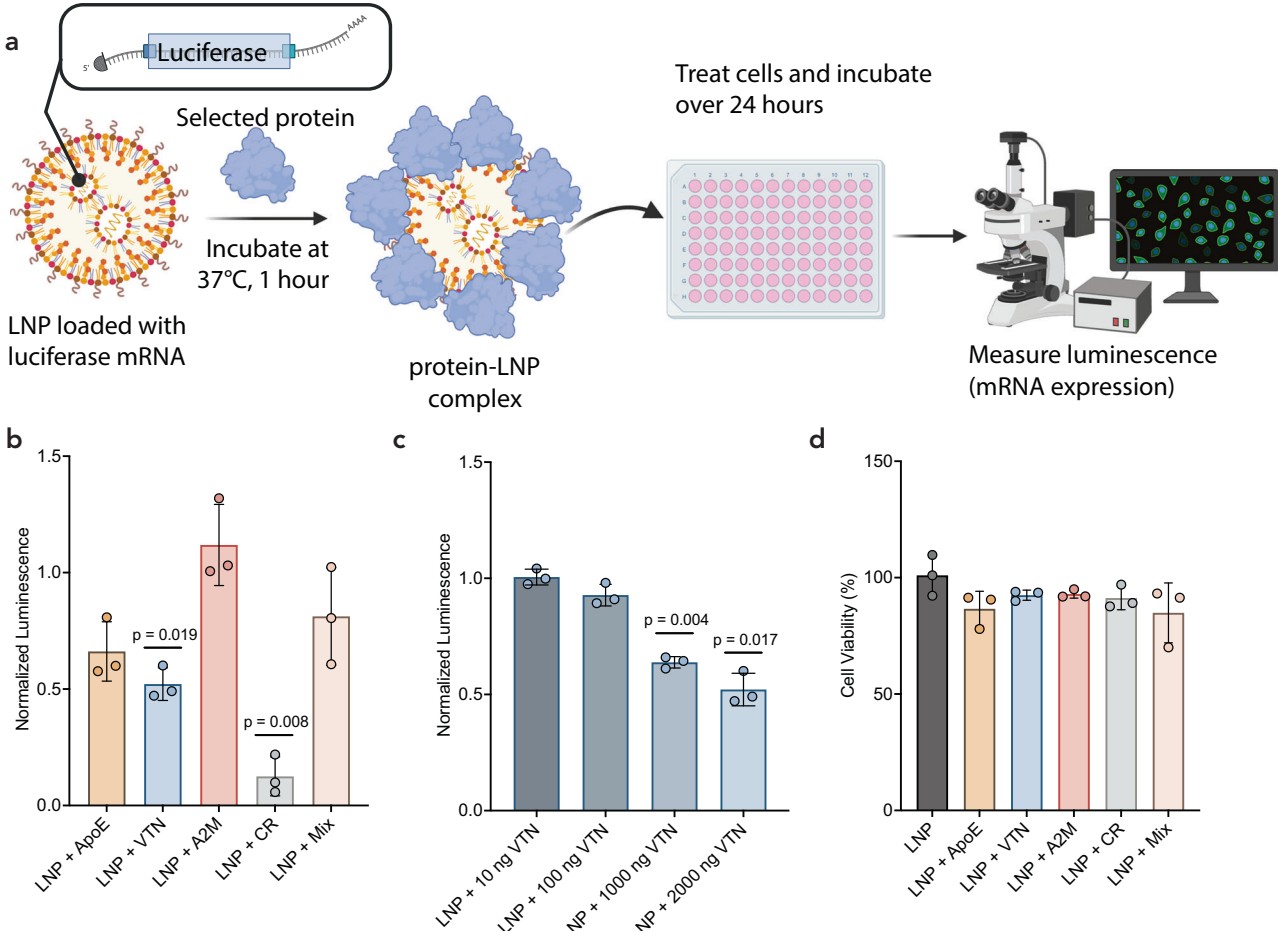

**Fig. 4 | In vitro mRNA expression with delivery by protein-LNP complexes. a** LNPs loaded with mRNA encoding luciferase were incubated with selected high-binding corona proteins (0.05 ng mRNA: 1 ng protein) prior to introduction to HepG2 cells seeded at $4.7 \times 10^4$ cells per cm² (100 ng mRNA per well). The luminescence was measured as a proxy for mRNA expression to understand the effect of proteins on LNP delivery efficiency. Luminescence was normalized to the average of each no-corona LNP biological control for all in vitro studies. Created in BioRender. Voke, E. (2025) https://BioRender.com/58u4s4t. **b** Resulting luminescence of pre-incubations of individual proteins with LNPs showed no significant change in luminescence (mRNA expression) for ApoE, A2M, or a mixture of the proteins, while showing a significant decrease for VTN or CR, each relative to the no-corona LNP control. **c** Dose–response of protein concentrations for VTN incubated LNPs showed a significant decrease in mRNA expression compared to the no-corona LNP control. **d** Cell viability showed no statistical difference for protein incubations. $N = 4$ technical replicates, $n = 3$ biological replicates. Data points shown are biological replicates. All data are presented as mean values with standard deviation. Statistical analysis was performed by repeated measures one-way ANOVA test with Geisser–Greenhouse correction, followed by Dunnett's multiple comparisons test where * and ** represent $p \le 0.05$ and $p \le 0.01$, respectively.

the membrane as a signal localized to the outer region of the cell. Through an erosion analysis of the cell within this focal plane, we studied the relative signal from the outer region of the cell, where LNPs may be stuck on the cell surface, and the inner region of the cell (Supplementary Fig. 9b, c). We calculated the fraction of Cy5 signal from the outer region relative to Cy5 signal from the entire cell (Supplementary Fig. 9d, e) and found that the LNPs incubated with ApoE, VTN, or a mixture of pre-formed coronas had more signal in this outer cell region compared to cells incubated with LNPs alone, though these differences were not statistically significant. Specifically, relative to the LNP-only cell incubation, cells incubated with VTN-LNPs or protein mixture-LNPs showed 8.5% and 9.4% increased Cy5 signal originating from the putative cell surface, respectively. These results suggest that the increased Cy5 signal associated with VTN-LNP or protein mixture-LNP incubated cells relative to LNP only incubated cells may be partially due to protein corona-induced LNP adhesion to the outer cell membrane.

To further validate differences in internalization for LNPs with pre-formed coronas observed during confocal microscopy, we used flow cytometry to measure cellular internalization of fluorescently labeled LNPs (Fig. 5d, e, Supplementary Fig. 11a, b). LNPs with and without a pre-formed corona were incubated with HepG2 cells for 1 hour at 37 °C before the cells were washed to remove LNPs on the outer surface. We found that uptake trends quantified with flow cytometry are consistent with the confocal microscopy data for both the percentage of Cy5-positive cells (Fig. 5d) and the difference in mean Cy5 fluorescence intensity between cells incubated with LNPs with a pre-formed protein corona compared to LNPs without a pre-formed corona (Fig. 5e). Specifically, we observe that cells exposed to LNPs pre-incubated with the protein mixture had four-fold higher levels of Cy5 mean fluorescence intensity than those exposed to LNPs without a pre-formed corona. Cells exposed to LNPs with an ApoE or VTN corona had a four-fold and 1.9-fold higher mean fluorescence intensity, respectively, though this difference was not statistically significant. In contrast, cells exposed to LNPs pre-incubated with A2M or CR had similarly low levels of mean fluorescence intensity as cells exposed to LNPs alone. These results provide an orthogonal method of validating our microscopy data, further supporting the aforementioned LNP internalization trends showing that certain protein coronas increase cellular uptake of LNPs. This counterintuitive result, that

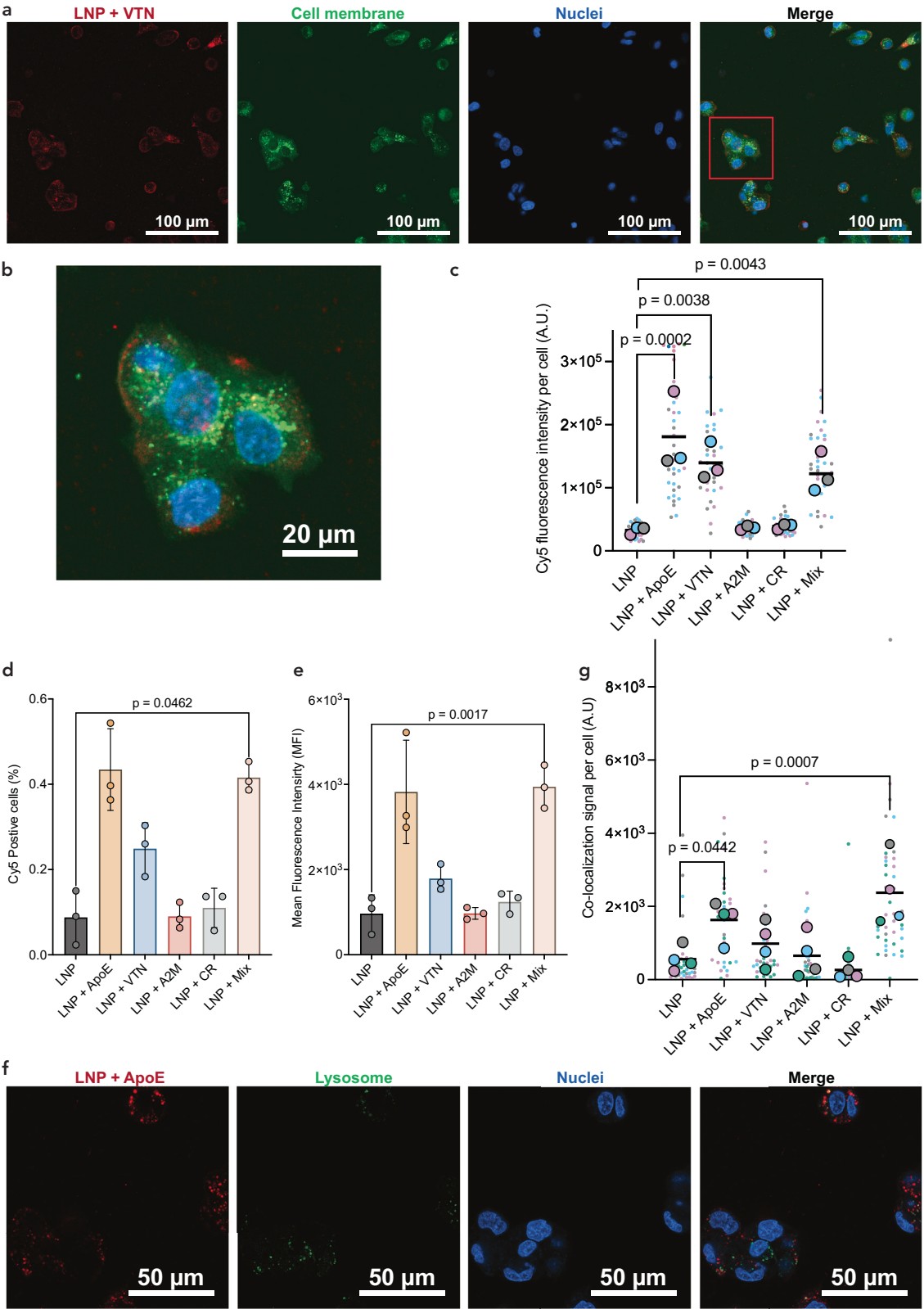

certain single-component pre-formed protein coronas and the select corona mixtures increase cell uptake while decreasing transfection efficiency, suggests that corona proteins may affect the efficiency of LNP endosomal escape.

Next, we considered whether pre-formed LNP coronas further affect LNP-cell interactions by influencing LNP endosomal escape, driving the observed discrepancies between LNP uptake and mRNA

expression. For effective cargo delivery, LNPs must escape the endosome before being trafficked to the lysosome for degradation[2]. To investigate differences between pre-formed coronas during intracellular trafficking, we compared the co-localization of lysosomes and LNPs. We quantified the co-localization of the Cy5-tagged LNP with the lysosome signal and normalized this signal per cell by the nuclei. We found significantly increased lysosomal co-localization for LNPs with

**Fig. 5 | Uptake and lysosomal co-localization of protein-LNP complexes in HepG2 cells. a** HepG2 cells internalizing LNPs loaded with Cy5-mRNA pre-incubated with high-binding corona proteins were visualized by confocal microscopy. Representative image of LNP + VTN incubations showing LNPs (Cy5; red), cell membrane (CellBrite membrane dye; green), and nuclei (Hoechst; blue). **b** Inset showing a magnified view of the region outlined by the red box in (**a**). **c** Quantification of Cy5 (LNP) signal per cell demonstrates differences in cell Cy5-mRNA uptake between select corona protein incubations ($n = 4$ technical replicates, $n = 3$ biological replicates). Each dot represents an individual field-of-view-level measurement, color-coded by biological replicate; larger, black-outlined dots indicate the mean value for each biological replicate. Statistical analysis was performed by a nested one-way ANOVA test followed by Dunnett's multiple comparisons test. **d, e** Cy5 signal from HepG2 cells internalizing LNPs loaded with Cy5-mRNA pre-incubated with high-binding corona proteins was also quantified by flow cytometry. **d** Percentage of Cy5-positive cells and **e** mean fluorescence intensity show uptake trends consistent with microscopy. Data points shown are biological

replicates ($n = 3$ technical replicates, $n = 3$ biological replicates). Error bars denote standard deviation. Statistical analysis was performed by repeated measures one-way ANOVA test with Geisser-Greenhouse correction, followed by Dunnett's multiple comparisons test. To compare endosome entrapment for select protein incubations, co-localization of the Cy5 signal (LNP) and fluorescently labeled lysosomes (green) was analyzed. Representative image of LNP + ApoE incubation shows **f** LNPs (red), lysosomes (green) and nuclei (blue) fluorescently labeled. **g** Quantification of overlapping Cy5 (LNP) and lysosome signal per cell ($n = 4$ technical replicates, $n = 4$ biological replicates). Each dot represents an individual field-of-view-level measurement, color-coded by biological replicate; larger, black-outlined dots indicate the mean value for each biological replicate. Statistical analysis was performed by a nested one-way ANOVA test followed by Dunnett's multiple comparisons test. For all statistical analyses performed, *, **, and *** represent $p \leq 0.05$, 0.01, and 0.001, respectively. All image thresholding was applied uniformly across samples, and image channels were adjusted solely for visualization purposes.

pre-formed ApoE or mixed protein coronas and no significant difference in lysosomal co-localization signal for LNPs with pre-formed VTN, A2M, or CR coronas (Fig. 5f–g). Specifically, we observe fourfold higher levels of LNP and lysosomal co-localization per cell for LNPs with a mixed protein corona compared to the LNPs without a pre-formed protein corona. We also found that LNPs with pre-formed ApoE coronas had approximately three-fold higher co-localization with lysosomes than the control protein-free LNP. Although we observe no statistically significant difference between the VTN-LNPs and LNPs without coronas, our data suggests that the LNPs with the pre-formed VTN corona have the next highest lysosomal colocalization signal. These results, in combination with the trends observed for the impact of protein coronas on mRNA expression, suggest that proteins influence LNP delivery efficiency at the level of both cell uptake and lysosomal trafficking.

## Discussion

In this work, we describe a workflow to characterize the protein corona on LNPs in a quantitative manner. We account for the presence of native particles in the biological fluid (here, blood plasma) through a continuous density gradient and abundance normalization. As informed by tracking the separation of fluorophore-tagged LNPs, we collect a subset of fractions from DGC that maximizes the concentration of LNPs and limits contamination from non-interacting proteins for proteomic analysis. We identify enriched LNP-corona proteins consistent with literature, such as apolipoproteins and vitronectin[28], as well as lower abundance proteins not previously identified within the LNP protein corona, such as C-reactive protein. We also detect only select apolipoproteins within the LNP protein corona, as demonstrated by the lack of highly abundant apolipoproteins ApoA-I and ApoA-II in our analysis.

Further analysis of enriched proteins revealed their functions as associated with lipid transport and cholesterol metabolism. The association of corona protein functions with both the innate and adaptive immune responses, as well as lipid transport and metabolism, is supported by previous work[24,51,62]. These observed functional associations with lipid transport align with the lipid composition of the LNPs, confirming interactions with proteins that are exchanged on lipoproteins during blood circulation. Additionally, identification of lipid-binding and immunoglobulin receptor-binding molecular functions suggests that we successfully isolated proteins that are biologically relevant to the LNP corona. Interestingly, gene ontology analysis also links seven enriched proteins to heparin binding, which may impact cell internalization, as seen with liposomes[63]. The discovery of proteins related to complement and coagulation cascades enriched in the corona phase is also in line with previous literature demonstrating that nanoparticles are often tagged for removal by the complement activation pathway[64].

We studied the impact of putative hard corona proteins on LNP functionality in vitro by comparing mRNA expression of LNPs pre-incubated with top corona proteins versus LNPs without a pre-formed corona. We found significantly decreased mRNA expression for LNPs incubated with VTN or CR proteins and no significant change in mRNA expression for LNPs incubated with ApoE, A2M, or protein mixtures. The decrease in mRNA expression for CR incubated LNPs is likely because CR, a protein secreted by the liver and associated with inflammation, activates the complement pathway and has a role in LNP destruction or clearance[65]. Conversely, VTN functions as a cell adhesion and spreading factor[66]. LNPs with a VTN-rich corona relative to ApoE have previously shown worse delivery outcomes in HepG2 cells[28]. Additionally, previous research has linked LNP formulations with specificity toward the lungs for mRNA expression with a VTN-rich corona[24]. Decreased mRNA expression in liver cells for LNP formulations with VTN-rich coronas aligns with our results, potentially enhancing the overall selectivity of LNPs toward other organs such as the lungs[24]. Our results, therefore, highlight the significance of understanding how protein-LNP interactions, and specifically the LNP protein corona, enhances or inhibits LNP cellular uptake and transfection of the mRNA cargo.

To understand the observed differences in mRNA expression for certain protein coronas, we compared the levels of cell uptake and lysosomal trafficking for LNPs pre-incubated with our selected proteins. Counterintuitively, we found that although LNPs incubated with VTN or CR displayed decreased levels of mRNA expression, they did not have decreased levels of cell uptake (Fig. 6a). In fact, VTN-LNPs showed increased cell uptake relative to LNPs not pre-incubated with protein, while LNPs incubated with CR had no significant difference in uptake relative to LNPs without a pre-formed corona according to our microscopy results. We hypothesized that this increase in cellular uptake for VTN-LNPs may be due to their association with the membrane rather than internalization into the cell cytoplasm, as our localization analysis supported the conclusion that VTN-LNPs generally adhere more to the outside of the cell relative to protein-free LNPs. Specifically, VTN-LNPs, when compared to LNPs incubated without protein, show 8.5% more signal localized to the outer region of the cell versus inside the cell, suggesting that the association with the outside of the cell may prevent effective cargo delivery, leading to decreased mRNA expression. To further investigate VTN-LNP uptake, we applied flow cytometry to quantify cell uptake of LNPs with pre-formed VTN coronas after a wash step, which removes LNPs bound to the outside of the cell. We found that although VTN-LNPs still show slightly more cell uptake than LNPs without a pre-formed corona, the difference between the two conditions is smaller and not statistically significant. Together, these results suggest that LNPs pre-incubated with VTN may exhibit lower mRNA expression, partially due to LNP adhesion to the outer cell membrane. Furthermore, VTN corona proteins may have an

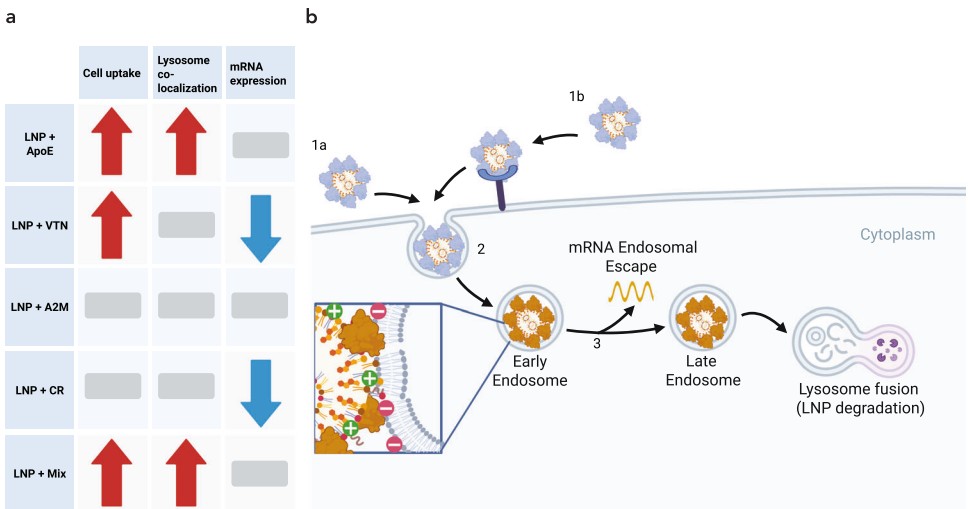

**Fig. 6 | Mismatch between mRNA expression and cell uptake. a** Differences in cell uptake, lysosome co-localization, and mRNA expression for protein-LNP complexes (arrows indicate increase or decrease and bars indicate no change). Created in BioRender. Voke, E. (2025) https://BioRender.com/58u4s4t. **b** Proteins influence LNP uptake into the cell through non-specific (**1a**) and/or receptor-mediated uptake (**1b**). These protein-LNP complexes enter (2) early endosomes, where the lower pH (pH ~5) environment ionizes the LNP and may impact the protein charge depending on its isoelectric point. Next, the mRNA must escape the endosome for protein expression (3). These proteins likely impact LNP endosomal escape, leading to different mRNA expression outcomes. Created in BioRender. Voke, E. (2025) https://BioRender.com/58u4s4t.

additional impact on LNP endosomal escape, as we observe a small but not statistically significant increase in lysosomal co-localization for LNPs with pre-formed VTN coronas (Fig. 5g).

The observed mRNA expression and uptake patterns for LNPs with pre-formed ApoE and mixed protein coronas also provide evidence that proteins influence LNP functionality beyond uptake. LNPs pre-incubated with ApoE had no significant increase in mRNA expression but had a five-fold or four-fold increase in cellular uptake in comparison to LNPs without pre-formed coronas, according to microscopy and flow cytometry, respectively. Similarly, LNPs pre-incubated with mixed proteins had no significant increase in mRNA expression but had a four-fold increase in cellular uptake in comparison to LNPs without pre-formed coronas, as demonstrated by both microscopy and flow cytometry. Co-localization analysis of ApoE-LNPs with lysosomes revealed a 2.9-fold increase in lysosome co-localization for LNPs with pre-formed ApoE coronas relative to LNPs alone, while LNPs with mixed protein corona had a four-fold increase in lysosomal co-localization. In both cases, we find a similar increase in both cell uptake and lysosome co-localization for LNPs with pre-formed ApoE and mixed protein coronas. Specifically, the four-fold increase in cell uptake and the 2.9-fold or four-fold increase in lysosomal co-localization suggest that, as more LNPs enter the cell, more LNPs are also trafficked to the lysosome for degradation. Lysosomal degradation of these LNPs likely accounts for the similar levels of mRNA expression between the ApoE- or mixed corona LNPs and the LNPs without a pre-formed corona. These results suggest that although the presence of an ApoE corona is beneficial for cell uptake into hepatocytes, the ApoE corona may also be inhibiting endosomal escape for this LNP. As these proteins enter the acidic environment of the early endosome, the net negative charge of ApoE shifts to a net positive charge with an isoelectric point of ~5.65, potentially influencing ApoE-lipid interactions and affecting endosomal escape[67,68].

Our results provide evidence that enriched LNP corona proteins influence mRNA cargo delivery beyond cellular internalization, suggesting a more complex mechanism of LNP endosomal escape[69]. Future therapeutic development requires further study of how protein-LNP interactions govern this key endosomal escape bottleneck in limiting LNP efficiency, yet our study highlights the contribution of the protein corona in hindering endosomal escape. Additionally, LNP design would benefit from considering how LNP corona-enriched proteins and/or their peptides that enhance cellular internalization may be leveraged to simultaneously mitigate trafficking to the lysosome. We can then design LNPs with favorable biomolecular interactions to optimize LNP function, as demonstrated recently by the development of a prototype apolipoprotein-based nanoparticle that leverages native lipoprotein trafficking as a delivery strategy[70]. Alternatively, strategies to prevent the interaction of specific proteins that activate inflammation could be considered. Collectively, these findings highlight that understanding the protein corona is important for the rational design of LNPs to overcome delivery bottlenecks at the points of cell entry and endosomal escape.

We highlight the importance of characterizing proteins with a high affinity for the LNP surface and provide a workflow that is easily adoptable for a wide range of soft nanoparticles, including liposomes, protein-based nanoparticles, and DNA nanostructures that fail to separate using conventional corona-isolation techniques. These historically understudied soft nanoparticles, which comprise 44% of nanoparticles in clinical trials, would benefit from further study of biomolecular interaction governing nanoparticle functionality using our workflow[71,72]. The quality-control measurements throughout the protocol enable extending our protocol to particles beyond LNPs and to biofluids beyond blood plasma.

Potential limitations are that the workflow isolates proteins with a high affinity for the LNP surface, known as the "hard corona", whereas proteins comprising the more transient and dynamic soft LNP corona may not be present after the isolation process. However, the separation protocol parameters could be further refined to retain soft corona proteins. Certain nanoparticles with low stability under shear forces or in biofluids of interest may not remain colloidally stable within the density gradient, underscoring the importance of the intermediate quality-control assessments that we outline. Additionally, denser nanoparticles, which pellet with tabletop centrifugation or incubations with biofluids that do not contain endogenous particles, would not benefit from this approach. Our approach is most suitable for identifying the proteins that most consistently and predominantly adhere to LNP surfaces, rather than weaker and transient protein-LNP interactions.

In summary, we provide new methods to quantify the LNP protein corona and its downstream effects on LNP mRNA delivery efficacy. We

found that select proteins distinctly influence LNP internalization, endosomal escape, and subsequent mRNA expression. These findings contribute to the growing evidence that biomolecular interactions heavily influence the mechanism of LNP delivery outcomes, shown here for mRNA delivery efficiency in cells, but likely also for additional outcome measures, including biodistribution, biocompatibility, stability, and in vivo efficacy. Further study is required to untangle the complexity of these protein-LNP interactions and their influence on the broad and growing range of clinical applications supported by LNP technologies. By understanding these protein-nanoparticle interactions, we can tune the design of future mRNA-based biotechnologies for improved translation to clinical practice.

## Methods

### Materials

Helper lipids (1,2-dioleoyl-sn-glycero-3-phosphoethanolamine: DOPE, 850725), C-14-PEG-2000 (880150 P), and Liss Rhod PE (810150) were purchased from Avanti Polar Lipids. Cholesterol (C8667) was purchased from Sigma-Aldrich. Cre recombinase and luciferase mRNA were acquired from Translate Bio, now Sanofi. EZ Cap™ Cy5 Firefly Luciferase mRNA (R1010) was purchased from ApexBio Technology. Thermo Scientific™ Slide-A-Lyzer™ dialysis cassettes (66330) were purchased from Thermo Fisher Scientific. Pooled human blood plasma (991-58-P-RC) was purchased from Lee BioSolutions. OptiPrep Density Gradient Medium (ab286850) was purchased from Abcam. Open-top polyclear 12-mL ultracentrifuge tubes (NC9863486) were purchased from Thermo Fisher Scientific. The 21-gauge needles (305167) were purchased from BD. Cholesterol assay kit (AB65390) and Human Apolipoprotein AI ELISA kit (ab108803) were purchased from Abcam. The protein detergent removal kit (1632130) was purchased from Bio-Rad. Trypsin/Lys-C Mix (V5073) was purchased from Promega. Amicon 0.5-mL 3-kDa (UFC5003) and 30-kDa (UFC503024) molecular weight cut-off (MWCO) centrifugal filters were purchased from Sigma-Aldrich. EZQ Protein Quantitation Kit (R33200) and Pierce Peptide Quantitation Kit (23290) were purchased from Thermo Fisher Scientific. *Escherichia coli* chaperone protein ClpB, Hi3 *E. coli* standard (186006012) was purchased from Waters. Recombinant human ApoE (AB280330), recombinant human vitronectin (ab217407), recombinant human C-reactive protein (ab167710), and native human alpha-2-macroglobulin (ab77935) were purchased from Abcam. Greiner white-bottom 96-well plates (655083) and PerkinElmer black, clear-bottom 96-well plate (6055300) were purchased. The Bright-Glo™ Luciferase Assay System kit (E2610) was purchased from Promega. The CyQUANT™ MTT Cell Viability Assay kit (V13154) was purchased from Thermo Fisher Scientific. Hoechst 33342 (H1399) was purchased from Fisher Scientific. CellBrite™ Cytoplasmic Membrane Labeling Kit (30021) was purchased from Biotium. Invitrogen™ LysoTracker™ Green DND-26 (L7526) was purchased from Fisher Scientific.

### LNP synthesis

LNPs were synthesized according to our previously published work[73]. Lipidoid ($306O_{10}$), helper lipids (DOPE), cholesterol, and C-14- poly(ethylene glycol)(PEG)-2000 were dissolved in reagent-grade ethanol at 10 mg/mL. The lipidoid, helper lipid, cholesterol, and PEG were mixed in a 35:16:46.5:2.5 molar ratio, respectively. Subsequently, the citrate buffer was added to the lipid solution in a 1:10 volumetric ratio. Cre recombinase or luciferase mRNA was dissolved in 10 mM sodium citrate buffer at 1 mg/mL. Cre recombinase mRNA was used in LNPs for protein corona composition experiments, and luciferase mRNA was used in LNPs for in vitro experiments. The lipid solution was added to the mRNA solution at a 10:1 lipidoid to mRNA mass ratio and mixed by pipetting. The solution was then diluted with an equal volume of PBS saline. Lastly, the LNPs were dialyzed against 2 L of PBS for 1 hour in 0.5-mL 3.5-kDa MWCO Thermo Scientific™ Slide-A-Lyzer™ dialysis cassettes. LNPs for protein corona isolation and in vitro studies were formulated at final mRNA concentrations of 0.05 and 0.01 mg/mL mRNA, respectively.

### Fluorescently tagged LNP synthesis

Fluorescently tagged LNPs were synthesized based on the standard LNP synthesis method described above, with the addition of 0.5 mol % fluorescently tagged lipid, 1,2-dioleoyl-sn-glycero-3-phosphoethanolamine-N- (lissamine rhodamine B sulfonyl) (Liss Rhod PE). The lipidoid, helper lipid, fluorescently tagged lipid, cholesterol, and PEG were mixed in a 35:15.5:0.5:46.5:2.5 molar ratio, respectively.

### Dynamic light scattering

Hydrodynamic size distribution of LNPs was determined in a 10-fold PBS dilution to a concentration of 0.005 mRNA mg/mL LNPs using DLS (Malvern ZetaSizer Nano, Malvern Instruments).

### Protein corona isolation

LNPs synthesized at a concentration of 0.05 mg/mL mRNA with Cre recombinase mRNA were incubated with an equal volume (400 µL) of pooled human blood plasma at 37 °C, the physiologically relevant temperature, for 1 hour, which has previously been determined as sufficient time for corona formation to occur[40]. Simultaneously, a PBS control was incubated with equal volume (400 µL) of pooled human blood plasma at 37 °C for 1 hour. Iodixanol solutions were prepared the same day and chilled on ice prior to gradient preparation according to protocols established for exosome purification[55]. Directly after incubation, each sample was diluted to a final concentration of 30% iodixanol (OptiPrep Density Gradient Medium) with a total volume of 2 mL and loaded into the bottom of a polyclear 12-mL ultracentrifuge tube. This bottom layer was followed by 2-mL layers of 25%, 20%, 15%, 10%, and 5% iodixanol, resulting in a six-layer iodixanol gradient. These layers were added to the tube with a 21-gauge needle beginning from the bottom layer to the top layer, proceeding slowly to avoid splashing/mixing of layers and avoiding the introduction of bubbles, which disrupt the gradient during centrifugation. The difference in density between each of the six gradient layers should be visible (Supplementary Fig. 12). Two tubes, one containing the LNPs incubated with plasma and one with a plasma control, were centrifuged for 16 hours at 36,000 rpm ($160,000 \times g$) and 4 °C with minimum acceleration and no braking in a SW 41 Ti Beckman swinging bucket rotor. Post centrifugation, 0.5-mL volume fractions were collected from the top to the bottom of the tube by careful pipetting. We added Triton-X 100 to the selected fractions as determined by the fluorescence assay to a final concentration of 2% Triton-X 100 to disrupt LNPs and then pooled them together using Amicon 0.5-mL 3-kDa MWCO centrifugal filters pre-rinsed with 50 mM Tris-HCl pH 8 at 4 °C according to manufacturer's instructions.

### Protein sample preparation for characterization

Following sample pooling, an acid-based protein precipitation method (Bio-Rad detergent removal kit) was used to remove ionic contaminants that interfere with LC-MS/MS, including detergents and free lipids. Further sample preparation followed our previously established protocols[32]. Proteins were reduced by heating at 37 °C for 60 min in urea/dithiothreitol (DTT) reducing buffer (8 M urea, 5 mM DTT, 50 mM Tris-HCl, pH 8). Proteins were alkylated with 15 mM iodoacetamide for 30 minutes in the dark. Next, 500 mM DTT was added to quench excess iodoacetamide in a volume ratio of 3:1 and incubated for 20 minutes. These samples were concentrated and filtered with 0.5-mL 3-kDa MWCO centrifugal filters pre-rinsed with 50 mM Tris-HCl pH 8. Protein concentration was determined with the EZQ Protein Quantitation Kit before 1:1 dilution with 50mM Tris-HCl, pH 8, to allow enzymatic protein digestion. In-solution protein digestion was done with a ratio of 1:25 weight/weight Trypsin/Lys-C (Mass Spectrometry Grade) to protein, overnight at 37 °C. Any remaining large

contaminants were removed by filtering with pre-rinsed Amicon 0.5-mL 30-kDa MWCO centrifugal filters. Peptide concentration was determined with the Pierce Peptide Quantitation Kit, and samples were then normalized to the same mass concentration. Peptide solutions were spiked with 50 fmol of *E. coli* housekeeping peptide (Hi3 *E. coli* Standard, Waters) per 5 μL sample volume to enable protein quantification. Digestion was stopped by freezing samples to −20 °C.

## Protein characterization via liquid chromatography-tandem mass spectrometry

Samples of proteolytically digested proteins were analyzed using a Synapt G2-Si ion mobility mass spectrometer that was equipped with a nanoelectrospray ionization source (Waters, Milford, MA). The Synapt G2-Si was connected in line with an Acquity M-class ultra-performance liquid chromatography system that was equipped with reversed-phase trapping (Symmetry C18, inner diameter: 180 μm, length: 20 mm, particle size: 5 μm, part number 186007496) and analytical (HSS T3, inner diameter: 75 μm, length: 150 mm, particle size: 1.8 μm, part number 186007473, Waters) columns. The mobile phase solvents were water and acetonitrile, both of which contained 0.1% formic acid and 0.01% difluoroacetic acid (volume/volume)[74]. Data-independent, ion mobility-enabled, high-definition mass spectra and tandem mass spectra were acquired using the positive ion mode[75–78]. Instrument control and data acquisition were performed using MassLynx software (version 4.1, Waters). Peptide and protein identification and quantification using a label-free approach were performed using Progenesis QI for Proteomics software (version 4.2, Waters Nonlinear Dynamics)[79,80]. *Escherichia coli* chaperone protein ClpB (accession P63284, Hi3 *E. coli* standard) was used as an internal standard for protein quantification. Data were searched against the human protein database to identify tryptic peptides using ion accounting as peptide identification method, trypsin as digest reagent allowing up to three missed tryptic cleavages, carbamidomethylcysteine as a fixed post-translational modification, methionine sulfoxide as a variable post-translational modification, a target false discovery rate of less than four percent, three or more fragment ions per peptide, seven or more fragment ions per protein, one or more peptides per protein, and a minimum score of four[81].

## Proteomic data analysis

Proteins were filtered for $q$ values (FDR-adjusted $p$ values) <0.05. Database for Annotation, Visualization and Integrated Discovery (DAVID) was used for functional annotation of gene ontology (GO), and Kyoto Encyclopedia of Genes and Genomes (KEGG) was used for pathway analysis of enriched proteins[82,83]. KEGG analysis relates known biological pathway maps to protein IDs of interest[84–86]. For GO and KEGG analysis using DAVID, the thresholds were based on the count (number of IDs) and EASE score (a modified Fisher Exact p-value for gene-enrichment analysis) which were set to 5 and 0.05, respectively.

## In vitro luciferase delivery

HepG2 cells were sourced from the University of California, Berkeley Cell Culture Facility. HepG2 cells were cultured in Eagle's Minimum Essential Medium with 10% fetal bovine serum (volume/volume) and 1% penicillin-streptomycin (volume/volume)[87]. Before plating, cells were washed with serum-free media and seeded into a white-bottom 96-well plate (surface area = 0.32 cm$^2$ per well) at a density of 15,000 cells per well. The cells were incubated at 37 °C for 24 hours in serum-free media. LNPs synthesized with luciferase mRNA at 0.01 mg/mL mRNA were incubated with proteins for 1 hour at 37 °C to a final LNP concentration of 0.005 mg/mL mRNA. The LNPs were incubated with 2000 ng of each protein (0.05 ng mRNA:1 ng protein, 0.01 mg/mL protein) unless otherwise specified. Following the incubation, each well was incubated with 20 μL of LNPs with or without the pre-formed protein corona at 0.005 mg/mL mRNA (100 ng mRNA per well) as

optimized previously[19]. After 24 hours, Brightglow Bright-Glo™ Luciferase Assay System kit and a plate reader were used to quantify mRNA expression via luminescence. Cell viability was also assessed at this time via the CyQUANT MTT assay. Endocytosis inhibition experiments were performed as described above with the addition of 50 μm Dynasore endocytosis inhibitor prior to LNP introduction to HepG2 cells.

## Confocal microscopy

HepG2 cells were cultured and plated according to conditions for the in vitro luciferase delivery assay. Cells were washed with serum-free media and were seeded into a black, clear-bottom 96-well plate at a density of 15,000 cells per well and incubated at 37 °C for 24 hours in serum-free media. LNPs were synthesized with EZ Cap™ Cy5 Firefly Luciferase mRNA at 0.01 mg/mL mRNA and incubated with proteins for 1 hour at 37 °C to a final LNP concentration of 0.005 mg/mL mRNA and 0.01 mg/mL protein concentration unless otherwise specified. Following the incubation, each well was incubated with 20 μL of LNPs with or without the pre-formed protein corona at 0.005 mg/mL mRNA (100 ng mRNA per well). For cell uptake experiments, 1.5 hours after LNP addition, cells were stained with Hoechst 33342 and CellBrite™ Cytoplasmic Membrane Labeling Kit. Images were acquired using a ZEISS Celldiscoverer 7 with $n = 3$ biological replicates and $n = 4$ technical replicates, with three fields of view (FOV) per technical replicate. Fields of view were collected in an unbiased automated fashion throughout each well at a focal plane offset of 4.5 μm from the bottom of the adherent cells using an air objective, ×20 (0.95) magnification, 0.5× tube lens, and a 43-second frame time. Images were collected with 0.8%, 0.1%, and 3% laser power for Cy5, CellBrite, and Hoechst, respectively. These acquisitions were taken by sequentially exciting Cy5 at 640 nm, CellBrite at 488 nm, and Hoechst at 405 nm. Emission was collected in the 617–700 nm range, 490–600 nm range, and 400–485 nm range, respectively. Images were batch processed by first creating a mask for the cell membrane based on the CellBrite dye. Then, the Cy5 signal within the mask was quantified and normalized according to the nuclei count per image. We calculated the fluorescence intensity as the summation of the Cy5 signal per FOV, with values from three FOVs mean-aggregated to a single technical replicate. For the erosion analysis, the inner membrane mask was acquired by eroding the membrane mask $n = 10$ times. The outer membrane mask was the exclusive disjunction of the total membrane mask and the inner membrane mask. Intensity was summed within each outer and inner mask, and the fraction was calculated based on Cy5 intensity within the membrane mask. For lysosomal colocalization of LNP analysis, 1.5 hours after LNP addition, cells were stained with Hoechst 33342 and Invitrogen™ LysoTracker™. Images were acquired using a ZEISS Celldiscoverer 7 with $n = 4$ biological replicates and $n = 4$ technical replicates, with three FOC per technical replicate. Fields of view were collected in an unbiased automated fashion throughout each well at a focal plane offset of 4.5 μm from the bottom of the adherent cells with a water immersion objective, ×50 (1.2) magnification, 0.5× tube lens, and a 34-second frame time. Images were collected with 0.8%, 0.2%, and 2% laser power for Cy5, Invitrogen™ LysoTracker™, and Hoechst, respectively. These acquisitions were taken by sequentially exciting Cy5 at 640 nm, Invitrogen™ LysoTracker™ at 488 nm, and Hoechst at 405 nm. Emission was collected in the 620–700 nm range, 490–602 nm range, and 400–495 nm range, respectively. Zen Blue 3.2 software was used for data collection. These images were batch processed by creating a mask for the lysosomes based on the Invitrogen™ LysoTracker™. Then, the Cy5 signal within the lysosome mask was quantified and normalized according to the nuclei count per image. We calculated the fluorescence intensity as the summation of the Cy5 signal per FOV, with values from three FOVs mean-aggregated to a single technical replicate. Further detailed analysis is available (https://github.com/tengjuilin/internalization-analysis). No statistical difference in cell count was observed for images between the LNP

control and pre-formed coronas in internalization and lysosomal co-localization experiments (Supplementary Fig. 13).

## Flow cytometry of LNP internalization

HepG2 cells were cultured and plated according to conditions for the in vitro luciferase delivery assay. Cells were washed with serum-free media and were seeded into a white-bottom 96-well plate at a density of 15,000 cells per well and incubated at 37 °C for 24 hours in serum-free media. LNPs were synthesized with EZ CapTM Cy5 Firefly Luciferase mRNA at 0.01 mg/mL mRNA and incubated with proteins for 1 hour at 37 °C to a final LNP concentration of 0.005 mg/mL mRNA and 0.01 mg/mL protein concentration unless otherwise specified. Following the incubation, each well was incubated with 20 μL of LNPs with or without the pre-formed protein corona at 0.005 mg/mL mRNA (100 ng mRNA per well). The cells were then incubated for 1 hour at 37 °C. The cells were then washed with PBS, removed from plates via trypsin digestion, and resuspended with FACS buffer (PBS, 2% FBS). Samples were acquired on a Thermo Fisher Attune NxT Acoustic Focusing Cytometer (Thermo Fisher Scientific), and data were analyzed using FlowJo v10 (FlowJo Inc) with the Cy5 mRNA positive gate set based on a control of non-treated cells (Supplementary Fig. 14).

## Statistics

Statistical analysis and visualization were performed with GraphPad Prism (v.10.2.3) and Python (v3). For in vitro studies, all samples had at least $n = 3$ biological replicates, and no statistical methods were used to predetermine sample size. Investigators were not blinded during the data analysis. See figure legends for full details of replicates, statistical testing, and significance. For proteomic analysis, see the methods section for full details on analysis and significance.

## Reporting summary

Further information on research design is available in the Nature Portfolio Reporting Summary linked to this article.

## Data availability

Source data are provided with the paper. The proteomic datasets generated during and analyzed during the current study are available in the MassIVE repository, [https://doi.org/10.25345/C5251FZ24, identifier: MSV000098245]. Source data are available for Fig. 5 and Supplementary Figs. 9, 10, and 13 in the associated source data file. Raw image data underlying these figures are not deposited due to large file sizes but are available from the corresponding author upon reasonable request. Requests will be fulfilled within approximately 3 weeks.

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

## Acknowledgements

Research was conducted with the California Institute for Quantitative Biosciences/College of Chemistry Mass Spectrometry Facility, which received NIH support (grant number 1S10OD020062-01), and the UC Berkeley Cell Culture Facility (RRID is SCR_017924). Confocal imaging experiments were conducted at the CRL Molecular Imaging Center (RRID:SCR_017852), supported by Helen Wills Neuroscience Institute. E.V. and H.J.S. were supported by the Department of Defense (DoD) through the National Defense Science & Engineering Graduate (NDSEG) Fellowship Program. M.L.A. was supported by a National Science Foundation (NSF) Graduate Research Fellowship (award number DGE1745016) and by a National Institutes of Health (NIH) F31 Fellowship (award number 1F31AG077874-01A1). R.C. was supported by the NSF Postdoctoral Research Fellowships in Biology (PRFB, award 2305663) and the Burroughs Wellcome Fund Postdoctoral Enrichment Program (PDEP). A.L. was supported by a National Science Foundation Graduate Research Fellowship and the Foundation for Food & Agriculture Research (FFAR) Fellows Program. R.L.P. acknowledges support from the Schmidt Science Fellows program in partnership with the Rhodes Trust and the Burroughs Wellcome Fund Career Award at the Scientific Interface (CASI). We acknowledge support from a Dreyfus Foundation award (MPL), the Philomathia Foundation (MPL), an NSF CAREER award 2046159 (MPL), a McKnight Foundation award (MPL), a Simons Foundation Award (MPL), a Moore Foundation Award (MPL), a Brain Foundation Award (MPL), a Heising-Simons Fellowship (MPL), and a Polymaths Award from Schmidt Sciences, LLC (MPL). MPL is a Chan Zuckerberg Biohub Investigator, and a Hellen Wills Neuroscience Institute Investigator.

## Author contributions

Conceptualization: E.V., R.L.P., and M.L.A.; project organization, resources, and funding acquisition, M.P.L. and K.A.W.: methodology and experiment design, E.V., M.L.A., R.L.P., A.T.I., and A.L.: experiments and characterizations, E.V., H.J.S., L.Z., M.L.A., and R.C.: data analysis and visualization, E.V. and T.J.L.: writing—original draft, E.V.: writing, review & editing, all co-authors. R.L.P., K.A.W., and M.P.L. jointly supervised this work, and these authors contributed equally.

## Competing interests

K.A.W. is an inventor on US patents 9,227,917 (2016) and 9,439,968 (2016) related to the LNPs described here and is a consultant for several companies translating nonviral RNA delivery systems. The remaining authors declare no competing interests.
