## [Transparent Peer Review file · Nature Communications]

Protein corona formed on lipid nanoparticles compromises delivery efficiency of mRNA cargo

Corresponding Author: Professor Markita Landry

Version 0:

Reviewer comments:

Reviewer #1

(Remarks to the Author)

This manuscript addresses a crucial aspect of lipid nanoparticles (LNPs) as non-viral delivery vehicles for RNA therapeutics, focusing on the formation of the protein corona and its impact on LNP function. The topic is highly relevant, particularly in the context of enhancing the potency and specificity of LNP-based RNA delivery, such as in clinical applications like mRNA vaccines.

On one hand, the authors employ a highly innovative, quantitative, label-free mass spectrometry-based proteomics workflow that effectively addresses several long-standing challenges in studying protein coronas formed on LNPs in biological environments. The study's ability to isolate and characterize LNP-protein complexes from human blood plasma using continuous density gradients is a particularly interesting contribution to the field. This method provides a solution to the inherent difficulties of distinguishing LNP-associated proteins from other endogenous particles, such as lipoproteins and extracellular vesicles.

However, despite the potential significance of the research, the manuscript contains multiple issues that must be addressed before it can be considered for publication. These concerns primarily relate to the novelty, clarity, experimental design, and interpretation of the data.

Major comments:

The manuscript lacks a clearly defined hypothesis. While it claims to explore how biofluid proteins modify LNP function, the specific research question being addressed is not sufficiently clear. The study proposes that the protein corona influences LNP uptake and mRNA transfection efficiency but does not articulate how this connects to a broader mechanistic understanding or the therapeutic implications. The manuscript would benefit from a clearer focus on specific research questions, rather than a broad, unfocused exploration of protein corona characterization.

The methodology for isolating the protein corona is not well-controlled. While the authors acknowledge the challenges of distinguishing LNPs from endogenous particles in biological fluids, they do not convincingly demonstrate how their isolation method eliminates contamination from lipoproteins or other nanoparticles that could interfere with the analysis. This weakens the validity of the proteomic analysis and raises concerns about the reliability of the identified proteins.

The identification of proteins associated with LNPs is limited to a small number of proteins. While the authors claim to have identified 56 proteins, only a subset of these is enriched in the corona, and this raises questions about the reproducibility and relevance of the data. Moreover, the absence of control experiments to validate the function of these proteins in the context of LNP uptake and mRNA transfection diminishes the interpretative value of the results.

The authors report a "mismatched relationship" between corona proteins and transfection efficiency. However, the interpretation of this result is vague. The claim that certain proteins "compromise LNP transfection efficiency" is not sufficiently substantiated with clear evidence. The lack of in-depth analysis regarding how protein-LNP interactions alter LNP behavior at a cellular level is a significant oversight.

While the authors discuss the challenges of protein corona isolation and the limitations of current methods, they do not adequately address the limitations inherent in their own approach. The reliance on a single LNP formulation and a limited set of biological samples (human plasma) reduces the generalizability of the findings. The manuscript would benefit from a more thorough discussion of the potential shortcomings of the methods and how these might influence the results.

The confocal assessment of LNP internalization by cells requires significant revision. First, the manuscript does not provide a detailed description of the image quantification procedure, and the link provided for further details is broken. Second, to discriminate between intracellular and extracellular space, the authors used CellBrite™ cytoplasmic membrane staining. However, the staining appears poor, with the dye being engulfed by cells and highlighting vesicular structures rather than accurately labeling cell membranes. These artifacts are well-known and necessitate optimization of the cell labeling

procedure. To accurately quantify LNP incorporation, 3D optical sectioning and reconstruction are essential. It is common in the literature that nanoparticle uptake studies fail to distinguish whether nanoparticles are localized inside cells or merely on the surface [1, 2]. It is important to distinguish the precise localization of nanoparticles, especially considering specific functional applications [3]. Therefore, 3D reconstruction is a “must-have” for proper quantification of LNP uptake.

The authors report quantification of Cy5 (LNP) signal per cell using ($n = 4$ technical replicates, $n = 3$ biological replicates). However, it is unclear how many individual cells were measured from each biological replicate. There are established guidelines for appropriate sampling in quantitative microscopy, which recommend that if there is only a subtle change (e.g., 25%) between two conditions, then 100 images may be required for each condition to ensure statistical confidence [4]. The manuscript should apply appropriate statistical methods for microscopy image quantification and follow commonly accepted guidelines for quantitative microscopy [5].

The manuscript lacks a description of the statistical analysis used for biological experiments. How did the authors determine the sample size? How was statistical power assessed? Guidelines for statistical power analysis can be found in recent literature (e.g., <https://www.nature.com/articles/492180a>; <https://www.nature.com/articles/nature.2013.14131>; <https://www.nature.com/articles/nrg3706>). Addressing these aspects will greatly enhance the manuscript's credibility and the robustness of the results.

References

1. Rennick, J.J., A.P.R. Johnston, and R.G. Parton, Key principles and methods for studying the endocytosis of biological and nanoparticle therapeutics. *Nature Nanotechnology*, 2021. 16: p. 266-276.
2. Donahue, N.D., H. Acar, and S. Wilhelm, Concepts of nanoparticle cellular uptake, intracellular trafficking, and kinetics in nanomedicine. *Advanced Drug Delivery Reviews*, 2019. 143: p. 68-96.
3. Shin, H., et al., Quantifying the level of nanoparticle uptake in mammalian cells using flow cytometry. *Nanoscale*, 2020. 12: p. 15743-15751.
4. Jonkman, J., et al., Tutorial: guidance for quantitative confocal microscopy. *Nature Protocols*, 2020. 15: p. 1585-1611.
5. Lord, S.J., et al., SuperPlots: Communicating reproducibility and variability in cell biology. *Journal of Cell Biology*, 2020. 219: p. e202001064.

Reviewer #2

(Remarks to the Author)

This manuscript by Voke et al. reports on the development of density gradient centrifugation methods optimized for the isolation of lipid nanoparticles used for RNA delivery from human plasma samples, to enable mass spectrometry-based analysis of the protein corona established on LNPs in contact with plasma. Using this optimized methodology, they identify 3 proteins, alpha-2-macroglobulin, C-reactive protein, and vitronectin, which are reliably enriched in the corona of a model LNP composition. Interesting, they show that while some of these proteins enhance LNP association with cells in vitro at the single cell level, they may inhibit actual transfection. For one of the corona proteins that inhibited transfection, vitronectin, the authors provide confocal imaging evidence that vitronectin-treated LNPs are trapped at the cell plasma membrane rather than being internalized. Further, ApoE-opsonized LNPs were found to show enhanced accumulation in lysosomes, with VTN-opsonized LNPs also exhibiting a trend toward lysosomal accumulation. This is an interesting and technically well executed study, that provides some interesting initial observations and an experimental roadmap for further exploration of how LNP composition impacts protein corona formation, and ultimately LNP efficacy.

Major comments:

1. It seemed confusing that the analysis of cholesterol in the linear density gradient was largely localized to a set of fractions distinct from where the LNPs were localized, when the LNPs themselves contain cholesterol. Does this just reflect a low sensitivity of the cholesterol assay for the levels of cholesterol in the LNPs themselves?
2. The authors comment on an enrichment of albumin in the LNP fractions when shorter centrifugation times were used. How does one know if this is an artifact vs. reflecting weakly-bound albumin that was part of the corona, but which becomes dislodged with longer centrifugation times? This seems like a point that merits discussion.
3. The authors mention exosomes as important contaminants and their similarity in composition to LNPs, but no experimental analysis was made (unless I somehow missed it) of whether the linear density gradient is removing exosomes. This should be addressed.
4. The authors take 3 proteins consistently enriched in the LNP corona across independent technical runs, as well as ApoE, and test the impact of incubating LNPs with isolated proteins or the combination of these 4 on in vitro transfection. It would be valuable for this experiment to include a verifying analysis using the density gradient method that the individual proteins do indeed adsorb to the LNPs when incubated as purified proteins, to rule out cooperative effects that make them bind to the LNP when incubated in whole plasma.

Reviewer #3

(Remarks to the Author)

This manuscript investigates the impact of the protein corona on lipid nanoparticle (LNP) uptake and mRNA transfection efficiency. The authors synthesize LNPs, characterize their corona composition, and assess how the most abundant proteins

influence transfection. While this is an interesting study, a major concern is that the findings are largely limited to HepG2 cells, with no validation in more physiologically relevant models such as primary human cells or in vivo systems. Without demonstrating that these effects are generalizable across different biological environments, the broader implications of the study remain uncertain. Additionally, the observation that protein corona formation impairs mRNA transfection is well established. Serum proteins are known to interfere with transfection, which is why serum-free conditions are commonly used. While the study reinforces this phenomenon, it does not provide novel mechanistic insights or propose innovative strategies to address these limitations. Further, the role of vitronectin and other proteins in endosomal escape and transfection efficiency remains insufficiently explored. Although vitronectin enhances LNP uptake, it paradoxically reduces mRNA transfection efficiency. However, no significant differences in lysosomal colocalization were observed, casting doubt on the proposed mechanism. Additional experiments are needed to clarify this finding.

Major Concerns:

- The study lacks crucial details on the quantities of LNPs and plasma used for corona formation, making it difficult to assess how accurately the conditions mimic in vivo environments. This omission weakens the study's biological relevance.
- The authors mention that most of the LNPs were localized in fraction 2-6 of the iodixanol gradient, whereas most of the endogenous particles were present in fractions 5-10. But this analysis was based solely on the amount of cholesterol, as it has been considered as the key lipoprotein constituent. This assumption needs to be backed by more quantification.
- The authors mention the current limitations for protein corona quantification on LNPs. Specifically, it focuses on two key points : firstly, it highlights the importance of excluding the nano-sized particles which are intrinsically present in the biological fluids, and secondly it also talks about the low density of the soft nanoparticles, which causes difficulty to isolate protein-LNP complex using centrifugation. The authors back their claim in Supplementary Fig - 1A, where the ultracentrifugation was performed for 2 hours at 160,000 relative centrifugal force. We acknowledge and appreciate the need to quantify protein corona in a robust manner, however we recognize that the unbound proteins can also be eluted by ultracentrifugation for lesser time (e.g. 30 minutes) at a much lesser rcf as it has been established and verified previously in other studies. This questions the novelty of the work and further raises concerns about the robustness of the study.
- The in vitro studies focused on the four most abundant proteins identified in the protein corona. While understanding the individual contributions of these proteins is valuable, a comprehensive assessment of the corona's impact on LNP functionality requires considering all the hard corona proteins.
- Internalization studies are based solely on imaging. Incorporating flow cytometry or other quantification methods would provide a more rigorous analysis.
- The findings indicate greater internalization of VTN+LNP compared to LNP without a corona. However, an unexpected discrepancy was observed with mRNA transfection, showing lower transfection efficiency for VTN+LNP than for LNP without a corona. This suggests that while vitronectin enhances cellular uptake, it may hinder endosomal escape, potentially leading to reduced mRNA transfection. However, no significant differences were observed in lysosomal colocalization between VTN+LNP and LNP without a corona, raising questions about this conclusion. Further studies are necessary to clarify how vitronectin influences endosomal escape and transfection efficiency.

Version 1:

Reviewer comments:

Reviewer #1

(Remarks to the Author)

The authors have done an excellent job addressing the revisions. I have no further comments.

Reviewer #3

(Remarks to the Author)

The authors have addressed my concerns, and I have no further comments.

Reviewer #4

(Remarks to the Author)

The authors have carried out substantial additional measurements in response to reviewer queries, and in my opinion have suitably addressed the concerns raised. This new data strengthens the conclusions of the original manuscript. The revised manuscript is much improved and I would recommend acceptance.

REVIEWER COMMENTS

We thank the reviewer for their thorough analysis of our manuscript and suggestions for the improvement of our study.

Reviewer #1 (Remarks to the Author):

This manuscript addresses a crucial aspect of lipid nanoparticles (LNPs) as non-viral delivery vehicles for RNA therapeutics, focusing on the formation of the protein corona and its impact on LNP function. The topic is highly relevant, particularly in the context of enhancing the potency and specificity of LNP-based RNA delivery, such as in clinical applications like mRNA vaccines.

On one hand, the authors employ a highly innovative, quantitative, label-free mass spectrometry-based proteomics workflow that effectively addresses several long-standing challenges in studying protein coronas formed on LNPs in biological environments. The study's ability to isolate and characterize LNP-protein complexes from human blood plasma using continuous density gradients is a particularly interesting contribution to the field. This method provides a solution to the inherent difficulties of distinguishing LNP-associated proteins from other endogenous particles, such as lipoproteins and extracellular vesicles.

We appreciate the reviewers' positive comments on the significance of our manuscript.

However, despite the potential significance of the research, the manuscript contains multiple issues that must be addressed before it can be considered for publication. These concerns primarily relate to the novelty, clarity, experimental design, and interpretation of the data.

We thank the reviewer for their comments and insightful suggestions, which we have addressed as outlined below.

Major comments:

The manuscript lacks a clearly defined hypothesis. While it claims to explore how biofluid proteins modify LNP function, the specific research question being addressed is not sufficiently clear. The study proposes that the protein corona influences LNP uptake and mRNA transfection efficiency but does not articulate how this connects to a broader mechanistic understanding or the therapeutic implications. The manuscript would benefit from a clearer focus on specific research questions, rather than a broad, unfocused exploration of protein corona characterization.

We thank the reviewer for their comments, and we agree that further explanation on our specific research questions would strengthen our study. Our study both 1) develops a new approach to identify hard corona proteins for LNPs and also 2) undertakes a mechanistic study of how these newly-identified proteins affect LNP function. As such, we have revised our introduction and discussion to more clearly outline the goals and underlying hypotheses of our study. These focus areas are denoted in the introduction in lines 71-84, "We hypothesize that protein corona formation impacts the core functions of LNPs: delivery localization, cell internalization, and endosomal escape, all of which are required for mRNA cargo delivery. In this work, we applied a quantitative, label-free mass spectrometry-based proteomics workflow that leverages continuous density gradients to probe the nano-bio interface of LNPs in human blood plasma. Our approach accounts for the presence of native particles in the proteomic analysis of the corona, without modification of the LNP formulation or surface. We provide clarity on best practices for

sample preparation to reproducibly collect highly enriched LNP corona proteins, and through this approach, consistently find proteins associated with lipid transport and metabolism enriched in the corona. We identify a small set of proteins that form the putative hard corona on LNPs and examine how they influence LNP transfection. By studying LNPs with pre-formed protein coronas, we discovered a mismatch between internalization and mRNA expression: certain corona proteins increased cellular uptake of LNPs by five-fold but had no effect on mRNA expression.”

We agree that further explanation on how our findings connect to broader mechanistic understanding would improve our manuscript. In lines 52-61, we comment on how our limited understanding of the connection between enriched proteins and biological outcomes may hinder the development of LNPs in the future. Furthermore, we've included more information on how corona characterization may guide LNP design in the introduction lines 86-93, “Connecting protein corona formation on LNPs to cellular delivery outcomes provides insight into the biomolecular mechanisms that limit LNP transfection, particularly the low endosomal escape efficiency estimated at 2%.² Our findings suggest that increased cellular uptake does not necessarily improve transfection, especially when the protein corona may hinder endosomal escape. We propose that the protein corona influences both LNP uptake and intracellular trafficking. However, pinpointing the specific proteins that are strongly and consistently enriched in the LNP corona remains experimentally challenging, limiting our ability to assess their influence on key steps of cargo delivery.

Additionally, we have added more discussion in lines 581-593 to specify how these findings can be more broadly applied, “Our results provide evidence that enriched LNP corona proteins influence mRNA cargo delivery beyond cellular internalization, suggesting a more complex mechanism of LNP endosomal escape.⁷⁰ Future therapeutic development requires further study of how protein-LNP interactions govern this key endosomal escape bottleneck in limiting LNP efficiency, yet our study highlights the contribution of the protein corona in hindering endosomal escape. Additionally, LNP design would benefit from considering how LNP corona enriched proteins and/or their peptides that enhance cellular internalization may be leveraged to simultaneously mitigate trafficking to the lysosome. We can then design LNPs with favorable biomolecular interactions to optimize LNP function as demonstrated recently by the development of a prototype apolipoprotein-based nanoparticle that leverages native lipoprotein trafficking as a delivery strategy.⁷¹ Alternatively, strategies to prevent the interaction of specific proteins that activate inflammation could be considered. Collectively, these findings highlight that understanding the protein corona is important for the rational design of LNPs to overcome delivery bottlenecks at the points of cell entry and endosomal escape.”

The methodology for isolating the protein corona is not well-controlled. While the authors acknowledge the challenges of distinguishing LNPs from endogenous particles in biological fluids, they do not convincingly demonstrate how their isolation method eliminates contamination from lipoproteins or other nanoparticles that could interfere with the analysis. This weakens the validity of the proteomic analysis and raises concerns about the reliability of the identified proteins.

We thank the reviewer for their comment, and we agree that, generally, full elimination of contaminants remains a challenge both in our work and also in the field. We highlight that our analysis - only considering corona proteins that are repeatedly enriched across numerous technical *and* experimental replicates - increases the chance that they are indeed LNP protein corona constituents. Furthermore, the downstream analysis showing that coronas pre-formed with these single constituent proteins significantly affect LNP efficacy further support the likelihood that they are not false positive findings. While we cannot - and do not - claim zero contamination from endogenous nanoparticles, we do experimentally show that

our method enables better LNP-specific corona isolation than prior research in this space, with minimal non-LNP protein corona contributions.

In our manuscript, we show that cholesterol, a constituent of lipoproteins, localizes to fractions 5-10, indicating that across all types of lipoproteins, their localization is minimized in the density gradient fractions from which we collect samples. Cholesterol is only present in human blood plasma within lipoproteins, and therefore we are confident that our approach minimizes contamination from lipoproteins. We nonetheless recognize the importance of orthogonally validating our claims that our method enables recovery of corona proteins predominantly from LNPs. Accordingly, we have performed additional control experiments to strengthen our analysis. We performed an enzyme-linked immunosorbent assay (ELISA) to quantify human apolipoprotein A-I within the density gradient as an example apolipoprotein. In this antibody-based approach, we confirm that the majority of apolipoprotein AI, 99.5%, is present in fractions 12-24, and that concentration of apolipoprotein AI is minimal in fractions 2-6. We added the figure to the Supplemental Information and updated the main text to reflect this additional data in lines 200-205, "To further validate the localization of lipoproteins, we quantified the presence of the most abundant apolipoprotein in human plasma, apolipoprotein AI (ApoA-I). We confirmed that 99.5% of ApoA-I localizes to fractions 12-24 through an enzyme-linked immunosorbent assay (ELISA; Supplementary Fig. 5)."

Supplementary Figure 5. Apolipoprotein AI localization within the density gradient. Average human apolipoprotein AI enzyme-linked immunosorbent assay quantification of plasma alone gradient fractions collected after DGC isolation workflow shows that 99.5% of apolipoprotein AI proteins are present among fractions 12-24. We estimated the percentage of lipoprotein A within the peak by calculating the area under the measured fluorescence curve for fractions 12-24 relative to the total area under the measured fluorescence curve using the trapezoidal rule (trapz function from scipy.integrate). N = 2 biological replicates. Error bars all denote standard deviation.

Additionally, as mentioned in lines 173-174, this protocol is based on methods used to separate subpopulations of exosomes. As such, the presence of exosomes within these density gradient conditions has been characterized in Temoche-Diaz et. al's work,¹ which shows exosome localization in their Figure 1b through a immunoblot of key exosome markers. In their work, 0.4 mL fractions are collected, while we collect 0.5 mL. They found no exosomes present in fractions 1-6 (equivalent to fractions 1-5 in our study) and the presence of low density exosomes in fractions 9-11 (equivalent to fractions 7-9 in our study). To clarify the position of exosomes within the density gradient, we have modified the manuscript text in lines 203-205 to "Previous work has already determined that exosomes, another type of endogenous particle, are not present in the first 5 fractions of the density gradient."⁵⁴

Figure 1b from Temoche-Diaz, M. M.; Shurtleff, M. J.; Nottingham, R. M.; Yao, J.; Fadadu, R. P.; Lambowitz, A. M.; Schekman, R. Distinct Mechanisms of microRNA Sorting into Cancer Cell-Derived Extracellular Vesicle Subtypes. *Elife* **2019**, *8*, e47544.

We are able to limit the presence of endogenous particles during isolation of LNPs in the density gradient as demonstrated with cholesterol and lipoprotein A quantification. We would also like to highlight that as illustrated in Supplementary Fig. 6, we only need to achieve partial separation of the endogenous particles to account for the presence of these endogenous particles in our proteomic analysis. By selecting for fractions maximizing concentrations of these LNPs, differences become clear in the proteomics downstream as we compare to a plasma alone control. The identification of select apolipoproteins but not the most abundant apolipoproteins supports that we have accounted for this contamination as discussed in lines 258-261, "Despite ApoA-I and apolipoprotein A-II (ApoA-II) being the two most abundant apolipoproteins in blood plasma,⁵⁸ we do not identify ApoA-I or ApoA-II as enriched in the protein corona, suggesting we are selectively identifying apolipoproteins that interact with LNPs."

The identification of proteins associated with LNPs is limited to a small number of proteins. While the authors claim to have identified 56 proteins, only a subset of these is enriched in the corona, and this raises questions about the reproducibility and relevance of the data.

We thank the reviewer for their comments. However, we disagree with the assumption that a small number of proteins indicates a limit to the reproducibility and relevance of the proteomic data; on the contrary, we argue that this supports the robustness and specificity of our findings. Specifically, we

believe that the reproducible identification of a small subset of proteins (across multiple independent technical and experimental replicates) increases the certainty that the identified proteins are enriched in the LNP corona, reducing the probability that we would identify them through proteomic analysis by chance. As seen with other robust corona isolation methods, we would expect only a small number of proteins to be reproducibly enriched in the corona. For instance, as seen in Pattipeiluhu and co-authors work, their photoaffinity method identified less than 20 corona proteins for 3 different liposome formulations.² Additionally, a recent magnetic-based separation approach found a similar number of 52 proteins enriched in the LNP protein corona.³ We have modified the text in lines 242-243 to clarify our argument, "Other label-based corona isolation methods have identified a similar number of proteins enriched in the LNP protein corona."^{47,49}

Moreover, the absence of control experiments to validate the function of these proteins in the context of LNP uptake and mRNA transfection diminishes the interpretative value of the results. The authors report a "mismatched relationship" between corona proteins and transfection efficiency. However, the interpretation of this result is vague. The claim that certain proteins "compromise LNP transfection efficiency" is not sufficiently substantiated with clear evidence. The lack of in-depth analysis regarding how protein-LNP interactions alter LNP behavior at a cellular level is a significant oversight.

We thank the reviewer, and we agree that additional control experiments are necessary to more thoroughly substantiate our claims. We further studied how these interactions alter LNP behavior by 1) validating that these protein-LNP complexes undergo endocytosis and 2) applying flow cytometry as an orthogonal technique to quantify LNP internalization.

To validate that these protein-LNP complexes undergo endocytosis, we investigated how the presence of endocytosis inhibitors would influence mRNA expression for LNPs pre-incubated with proteins. We confirmed that endocytosis inhibitors prevent mRNA expression, suggesting that differences in mRNA expression beyond uptake into cells are likely driven by differences in endocytosis escape. We have modified the text in lines 374-382, "We also confirmed that LNP uptake occurs through the anticipated pathway of endocytosis. To test this, we measured mRNA expression in HepG2 human liver cells treated with Dynasore and LNPs that were either bare or protein corona-coated. Dynasore functions as an inhibitor of clathrin-coated pit-mediated endocytosis (CME) as well as fast endophilin-mediated endocytosis (FEME), a dynamin-dependent, clathrin-independent pathway for rapid ligand-driven endocytosis."⁶⁴ We found that mRNA expression from LNPs both with and without the pre-incubated corona were entirely reliant on dynamin-dependent endocytosis pathways (Supplementary Fig. 8a). These effects were observed at inhibitor concentrations that did not influence cell viability (Supplementary Fig. 8b)."

Supplementary Figure 8. Endocytosis inhibition of HepG2 cells incubated with protein-LNP complexes. LNPs loaded with mRNA encoding luciferase were incubated with selected high-binding corona proteins (0.05 ng mRNA : 1 ng protein) prior to introduction to HepG2 cells seeded at 4.7×10^4 cells per cm^2 (100 ng mRNA per well) pre-incubated for 30 minutes with 50 M Dynasore endocytosis inhibitor. The luminescence was measured as a proxy for mRNA expression to understand the effect of proteins on LNP delivery efficiency in the presence of endocytosis inhibition. (a) Resulting luminescence of pre-incubations of individual proteins with LNPs showed a significant decrease in luminescence (mRNA expression) for all conditions with the inhibitor. (b) Cell viability showed no statistical difference for inhibitor incubations. N = 4 technical replicates, n = 3 biological replicates. Data points shown are biological replicates. Error bars all denote standard deviation. Statistical analysis was performed by one-way ANOVA test followed by Dunnett's multiple comparisons test where * represent $p \leq 0.05$.

We applied flow cytometry to assess whether the trends we observed during confocal microscopy hold. Indeed, we find that the trends we observed during confocal microscopy image analysis are consistent with the trends we observed from flow cytometry. We have modified Figure 5 and the manuscript to include this supporting data in lines 465-482, "To further validate differences in internalization for LNPs with pre-formed coronas observed during confocal microscopy, we used flow cytometry to measure cellular internalization of fluorescently labeled LNPs (Fig. 5d-e, Supplementary Fig. 11a-b). LNPs with and without a pre-formed corona were incubated with HepG2 cells for 1 hour at 37 °C before cells were washed to remove LNPs on the outer surface. We found that uptake trends quantified with flow cytometry are consistent with the confocal microscopy data for both the percentage of Cy5 positive cells (Fig. 5d) and the difference in mean Cy5 fluorescence intensity between cells incubated with LNPs with a pre-formed protein corona compared to LNPs without a preformed corona (Fig. 5e). Specifically, we observe that cells exposed to LNPs pre-incubated with the protein mixture had four-fold higher levels of Cy5 mean fluorescence intensity than those exposed to LNPs without a preformed corona. Cells exposed to LNPs with an ApoE or VTN corona had a four-fold and 1.9-fold higher mean fluorescence intensity, respectively, though this difference was not statistically significant. In contrast, cells exposed to LNPs pre-incubated with A2M or CR had similarly low levels of mean fluorescence intensity as cells exposed to LNPs alone. These results provide an orthogonal method of validating our microscopy data, further supporting aforementioned LNP internalization trends showing that certain protein coronas increase cellular uptake of LNPs. This counterintuitive result that certain single-component pre-formed protein

coronas and the select corona mixtures increase cell uptake while decreasing transfection efficiency suggests that corona proteins may affect the efficiency of LNP endosomal escape.”

While the authors discuss the challenges of protein corona isolation and the limitations of current methods, they do not adequately address the limitations inherent in their own approach. The reliance on a single LNP formulation and a limited set of biological samples (human plasma) reduces the generalizability of the findings. The manuscript would benefit from a more thorough discussion of the potential shortcomings of the methods and how these might influence the results.

We thank the reviewer for their insightful comments, and we agree that additional discussion of the limitations of our approach is needed. We have expanded our discussion in lines 618-620, “However, this workflow is limited to probing proteins with a high affinity for the LNP surface, known as the “hard corona”, whereas proteins comprising the more transient and dynamic soft LNP corona remain to be characterized.” The discussion has been modified to include more guidance on potential shortcomings in lines 611-620, “Potential limitations are that the workflow isolates proteins with a high affinity for the LNP surface, known as the “hard corona”, whereas proteins comprising the more transient and dynamic soft LNP corona may not be present after the isolation process. However, the separation protocol parameters could be further refined to retain soft corona proteins. Certain nanoparticles with low stability under shear forces or in biofluids of interest may not remain colloidally stable within the density gradient, underscoring the importance of the intermediate quality-control assessments that we outline. Additionally, denser nanoparticles, which pellet with tabletop centrifugation, or incubations with biofluids that do not contain endogenous particles would not benefit from this approach. Our approach is most suitable for identifying the proteins that most consistently and predominately adhere to LNP surfaces, rather than weaker and transient protein-LNP interactions.”

The confocal assessment of LNP internalization by cells requires significant revision. First, the manuscript does not provide a detailed description of the image quantification procedure, and the link provided for further details is broken.

We thank the reviewer for their comments. We apologize for the missing link to the quantification procedure which provides a detailed view into our image processing approach. We have made this accessible.

Second, to discriminate between intracellular and extracellular space, the authors used CellBrite™ cytoplasmic membrane staining. However, the staining appears poor, with the dye being engulfed by cells and highlighting vesicular structures rather than accurately labeling cell membranes. These artifacts are well-known and necessitate optimization of the cell labeling procedure.

The cells are readily identifiable through the image processing mask of the cell outline, and thus further optimization of the staining process is not necessary for cellular quantification specifically. Moreover, our addition of flow cytometry experiments, as updated in Figure 5, confirm internalization trends observed during confocal microscopy, providing validation of this experimental workflow.

To accurately quantify LNP incorporation, 3D optical sectioning and reconstruction are essential. It is common in the literature that nanoparticle uptake studies fail to distinguish whether nanoparticles are localized inside cells or merely on the surface [1, 2]. It is important to distinguish the precise localization of nanoparticles, especially considering specific functional applications [3]. Therefore, 3D reconstruction is a “must-have” for proper quantification of LNP uptake.

We thank the reviewer for their comments, and we appreciate that a more robust approach to quantify LNP internalization would be through 3D reconstruction. However, 3D reconstruction of LNPs entering into individual cells is too time intensive to compare across the multiple protein conditions with statistical confidence. To achieve 3D reconstruction of cells, we would need to collect ~50-70 images per cell with a sample size of ~500 cells per condition and an average of 20 cells per field of view. This would require a minimum of 1250 images per condition, which is not feasible given the experimental time window of 3D image collection.

Alternatively, we have collected images of HepG2 cells at an intermediate distance within the cell, 4.5 μm above the bottom of the cell. The projected cell cross section below from HepG2 cells incubated with VTN-LNPs under the same experimental conditions for the lysosomal co-localization used in Figure 5 shows that LNPs are visible within the cell. This example projection has been added to the Supplemental Information as Supplementary Figure 10.

a

b

Supplementary Figure 10. Lysosomal co-localization of protein-LNP complexes in HepG2 cells. (a-b) HepG2 cells internalizing LNPs loaded with Cy5-mRNA incubated with VTN corona proteins were visualized by confocal microscopy. LNPs labeled with Cy5 (red), lysosomes (green), and nuclei (blue) were fluorescently imaged. Imaris-rendered cross-section view of Z-stack at 4.5 μ m offset from bottom of cells illustrates that the LNPs are within the cell. A total of 50 Z-slices spanning a 20 μ m depth from the bottom to the top of the cell were collected for cross-sectional rendering.

Furthermore, we have added additional flow cytometry data to provide supporting evidence for the trends we observed with confocal quantification as updated in Figure 5.

The authors report quantification of Cy5 (LNP) signal per cell using ($n = 4$ technical replicates, $n = 3$ biological replicates). However, it is unclear how many individual cells were measured from each biological replicate. There are established guidelines for appropriate sampling in quantitative microscopy, which recommend that if there is only a subtle change (e.g., 25%) between two conditions, then 100 images may be required for each condition to ensure statistical confidence [4]. The manuscript should apply appropriate statistical methods for microscopy image quantification and follow commonly accepted guidelines for quantitative microscopy [5].

We thank the reviewer for their insightful comments, and we have updated the manuscript to clearly reflect the number of individual cells measured for each biological replicate for both the cell uptake and lysosomal co-localization analysis in Supplementary Figure 13. We have also modified the manuscript in lines 807-809, "No statistical difference in cell count was observed for images between the LNP control and pre-formed coronas in internalization and lysosomal co-localization experiments (Supplementary Fig. 13)." These cell counts demonstrate that there is no statistically significant difference in cell counts per condition for both internalization and lysosomal co-localization analyses. According to recently published literature, established protocols recommend 5 fields of view per condition for typical LNP internalization studies.⁴ As seen in our Supplemental Information in Figure 13, we are above the recommended number of fields of view with 4 technical replicates and 3 fields of view per replicate, which totals 12 fields of view per condition for each biological replicate. For our internalization analysis, across all conditions, we observe an average of 23 cells per field of view with a total sample size of 4953 cells with 20x magnification. For our lysosomal colocalization analysis, which were imaged at a higher magnification of 50x, we observe an average of 12.5 cells per field of view with a total sample size of 3592 cells across all conditions. Furthermore, we have improved our data visualization for confocal data to reflect the recommended guidelines for image quantification as seen in the updated Figure 5, which now provides information on the individual scenes as well as the summary replicate data.

Supplementary Figure 13. Average cell count per image for confocal microscopy experiments. (a) Cell count per image for HepG2 cells internalizing LNPs loaded with Cy5-mRNA incubated with high-binding corona proteins. Across all conditions an average of 23 cells per image were detected. No significant differences were observed between LNPs pre-incubated with and without proteins. (b) Cell count per image for lysosomal colocalization analysis. Across all conditions an average of 12.5 cells per image were detected. No significant differences between LNPs pre-incubated with or without proteins. Each dot represents an individual scene-level measurement, color-coded by biological replicate; black-outlined dots show biological replicate means. Statistical analysis was performed by a nested one-way ANOVA test followed by Dunnett's multiple comparisons test.

The manuscript lacks a description of the statistical analysis used for biological experiments. How did the authors determine the sample size? How was statistical power assessed? Guidelines for statistical power analysis can be found in recent literature (e.g., <https://www.nature.com/articles/492180a>; <https://www.nature.com/articles/nature.2013.14131>; <https://www.nature.com/articles/nrg3706>). Addressing these aspects will greatly enhance the manuscript's credibility and the robustness of the results.

We thank the reviewer for their comments, and we agree that our manuscript requires more description of our statistical analysis used to develop the study. In our manuscript we provided information on each statistical analysis in the Figure caption. For example, in lines 346-349 we specify that "Error bars all denote standard deviation. Statistical analysis was performed by repeated measures one-way ANOVA test with Geisser-Greenhouse correction, followed by Dunnett's multiple comparisons test where * and ** represent $p \leq 0.05$ and $p \leq 0.01$ respectively." We updated the statistics section of our work to reflect the reviewer's comments by including information on sample size, statistical power, and statistical analysis. We have modified this section in lines 827-830, "Statistical analysis and visualization were performed with GraphPad Prism (v.10.2.3) and Python (v3). For *in vitro* studies, all samples had at least $n = 3$ biological replicates, and no statistical methods were used to predetermine sample size. Investigators were not blinded during the data analysis. See figure legends for full details of replicates, statistical testing, and significance. For proteomic analysis, see methods section for full details on analysis and significance."

Additionally, we added more information in Figure captions throughout the manuscript to be more descriptive. In Figure 2, lines 223-227 have been modified, “Excitation/emission wavelengths of 560/580 nm were used to detect lissamine rhodamine-tagged LNPs. **N = 4 technical replicates, n = 3 biological replicates.** (d) Average total cholesterol quantification of **plasma alone gradient fractions** collected after DGC isolation workflow show that lipoproteins **within plasma** are present primarily among fractions 5-10 (dotted lines). **N = 2 technical replicates, n = 2 biological replicates. Error bars all denote standard deviation.**” In Figure 4, in lines 346-349, we provided more information on the one-way ANOVA such as not assuming equal standard deviations with the Geisser-Greenhouse correction, **“Statistical analysis was performed by repeated measures one-way ANOVA test with Geisser-Greenhouse correction, followed by Dunnett’s multiple comparisons test** where * and ** represent $p \leq 0.05$ and $p \leq 0.01$ respectively.”

We thank the reviewer for their insightful comments and interest in power analysis. We agree that a power analysis is an important consideration in many experiments, specifically for *in vivo* experiments and clinical trials. Our work focuses on *in vitro* characterization and in these types of characterization approaches, $n = 3$ biological experiments and 4 technical replicates are generally accepted in literature.⁵ However, we nonetheless performed a post-hoc power analysis using Gpower software to demonstrate that we have sufficient power for our analysis for internalization as an example. We had 3 biological replicates per 2 conditions for this analysis for a total sample size of 6. We assumed the Alpha error prob to be 0.05 and that we were only considering 2 groups because in our analysis we only compare between the control and the individual protein group. We find that we have a relatively large effect size for these observations. We find that in both these cases we have sufficient power based on our analysis (<0.80) as shown below.

Example data

	LNP	LNP + VTN	LNP + ApoE
Rep 1	25679	128055	252939
Rep 2	35774	173393	147160
Rep 3	36564	116874	142957
Average	32672	139441	181019
Standard Deviation	6069	29930	62320
Pooled standard deviation		15269.75	31307.56
Effect size f		1.56	1.8
Power		0.812	0.902

References

1. Rennick, J.J., A.P.R. Johnston, and R.G. Parton, Key principles and methods for studying the endocytosis of biological and nanoparticle therapeutics. *Nature Nanotechnology*, 2021. 16: p. 266-276.

2. Donahue, N.D., H. Acar, and S. Wilhelm, Concepts of nanoparticle cellular uptake, intracellular trafficking, and kinetics in nanomedicine. *Advanced Drug Delivery Reviews*, 2019. 143: p. 68-96.
3. Shin, H., et al., Quantifying the level of nanoparticle uptake in mammalian cells using flow cytometry. *Nanoscale*, 2020. 12: p. 15743-15751.
4. Jonkman, J., et al., Tutorial: guidance for quantitative confocal microscopy. *Nature Protocols*, 2020. 15: p. 1585-1611.
5. Lord, S.J., et al., SuperPlots: Communicating reproducibility and variability in cell biology. *Journal of Cell Biology*, 2020. 219: p. e202001064.

Reviewer #2 (Remarks to the Author):

This manuscript by Voke et al. reports on the development of density gradient centrifugation methods optimized for the isolation of lipid nanoparticles used for RNA delivery from human plasma samples, to enable mass spectrometry-based analysis of the protein corona established on LNPs in contact with plasma. Using this optimized methodology, they identify 3 proteins, alpha-2-macroglobulin, C-reactive protein, and vitronectin, which are reliably enriched in the corona of a model LNP composition. Interesting, they show that while some of these proteins enhance LNP association with cells in vitro at the single cell level, they may inhibit actual transfection. For one of the corona proteins that inhibited transfection, vitronectin, the authors provide confocal imaging evidence that vitronectin-treated LNPs are trapped at the cell plasma membrane rather than being internalized. Further, ApoE-opsonized LNPs were found to show enhanced accumulation in lysosomes, with VTN-opsonized LNPs also exhibiting a trend toward lysosomal accumulation. This is an interesting and technically well executed study, that provides some interesting initial observations and an experimental roadmap for further exploration of how LNP composition impacts protein corona formation, and ultimately LNP efficacy.

We sincerely appreciate the reviewer's positive assessment of our study and their recognition of its technical execution and potential impact on understanding LNP-protein corona interactions.

Major comments:

1. It seemed confusing that the analysis of cholesterol in the linear density gradient was largely localized to a set of fractions distinct from where the LNPs were localized, when the LNPs themselves contain cholesterol. Does this just reflect a low sensitivity of the cholesterol assay for the levels of cholesterol in the LNPs themselves?

We thank the reviewer for their comments, and we would like to clarify that this does not reflect the sensitivity of the cholesterol assay. The gradient used to quantify the presence of cholesterol did not contain LNPs. The gradient used to quantify the presence of cholesterol contained only human blood plasma so that we would only quantify the presence of cholesterol within lipoproteins. We have clarified this in the figure caption in line 225 to read "Average total cholesterol quantification of plasma alone gradient fractions collected after DGC isolation workflow show that lipoproteins within plasma are present primarily among fractions 5-10 (dotted lines)".

2. The authors comment on an enrichment of albumin in the LNP fractions when shorter centrifugation times were used. How does one know if this is an artifact vs. reflecting weakly-bound albumin that was part of the corona, but which becomes dislodged with longer centrifugation times? This seems like a point that merits discussion.

We thank the reviewer for their comment. We find high levels of serum albumin, the most abundant human plasma protein, in the plasma control sample, which does not contain any LNPs, within the fractions we selected for isolating LNPs at shorter time scales in Supplemental Table 2. The relatively high abundance of serum albumin in these fractions illustrates that at shorter time scales, more dense proteins do not fully sediment to the bottom even in the plasma control, indicating poor density separation of highly abundant proteins.

However, we appreciate that we may only be examining more strongly bound proteins which is mentioned as both a potential advantage or limitation to our technique. The discussion has been modified to further highlight that our approach enables quantification of hard corona proteins in lines 611-620. "Potential limitations are that the workflow isolates proteins with a high affinity for the LNP surface, known as the "hard corona", whereas proteins comprising the more transient and dynamic soft LNP corona may not be present after the isolation process. However, the separation protocol parameters could be further refined to retain soft corona proteins. Certain nanoparticles with low stability under shear forces or in biofluids of interest may not remain colloidally stable within the density gradient, underscoring the importance of the intermediate quality-control assessments that we outline. Additionally, denser nanoparticles, which pellet with tabletop centrifugation, or incubations with biofluids that do not contain endogenous particles would not benefit from this approach. Our approach is most suitable for identifying the proteins that most consistently and predominately adhere to LNP surfaces, rather than weaker and transient protein-LNP interactions."

3. The authors mention exosomes as important contaminants and their similarity in composition to LNPs, but no experimental analysis was made (unless I somehow missed it) of whether the linear density gradient is removing exosomes. This should be addressed.

We thank the reviewer for this comment and take this opportunity to clarify our protocol. As mentioned in manuscript lines 173-174, our protocol is based on methods used to separate subpopulations of exosomes. As such, the presence of exosomes within these density gradient conditions has been characterized in Temoche-Diaz et. al's work,¹ which shows exosome localization in Figure 1b through a immunoblot of key exosome markers. In their work, 0.4 mL fractions are collected, while we collect 0.5 mL. They found no exosomes present in fractions 1-6 (equivalent to fractions 1-5 in our study) and the presence of low density exosomes in fractions 9-11 (equivalent to fractions 7-9 in our study). To clarify the position of exosomes within the density gradient, we have modified the manuscript text in lines 203-205 to "Previous work has already determined that exosomes, another type of endogenous particle, are not present in the first 5 fractions of the density gradient."⁵⁴

Figure 1b from Temoche-Diaz, M. M.; Shurtleff, M. J.; Nottingham, R. M.; Yao, J.; Fadadu, R. P.; Lambowitz, A. M.; Schekman, R. Distinct Mechanisms of microRNA Sorting into Cancer Cell-Derived Extracellular Vesicle Subtypes. *Elife* 2019, 8, e47544.

4. The authors take 3 proteins consistently enriched in the LNP corona across independent technical runs, as well as ApoE, and test the impact of incubating LNPs with isolated proteins or the combination of these 4 on *in vitro* transfection. It would be valuable for this experiment to include a verifying analysis using the density gradient method that the individual proteins do indeed adsorb to the LNPs when incubated as purified proteins, to rule out cooperative effects that make them bind to the LNP when incubated in whole plasma.

We thank the reviewer for this insightful comment, and we agree that additional verification to show that individual purified proteins are enriched in the LNP corona strengthens our manuscript. In order to confirm that these individual proteins interact with the LNP, we mimicked the protein pre-incubation conditions for LNPs used for *in vitro* experiments and characterized the size change for the hydrodynamic radii of the LNPs using dynamic light scattering (DLS). DLS is commonly used to characterize protein corona formation on nanoparticles.^{6,7}

We found that incubating LNPs with these individual proteins indeed does increase the hydrodynamic radii of the LNPs, suggesting that these individual proteins form an associated LNP protein corona. We modified the manuscript text to reflect this additional information in lines 364-366 to "We measured the increase in the hydrodynamic radii of the LNPs after the protein incubations, confirming that these select proteins form an associated LNP corona (Supplementary Table 6) prior to their introduction to cells."

Supplementary Table 6. Dynamic light scattering intensity mean size and polydispersity index (PDI) of LNPs incubated with proteins for 1 hour at 37 °C

	Intensity Mean Size (nm)	Intensity Mean Size STDV	PDI	PDI STDV
LNP	129	1.9	0.101	0.005
LNP + ApoE	139	3.1	0.107	0.027
LNP + VTN	136	1.2	0.116	0.006
LNP + A2M	142	7.2	0.171	0.030
LNP + CR	188	0.8	0.094	0.002
LNP + Mix	168	29.3	0.167	0.016

As shown in Supplementary Table 6, LNPs incubated with individual proteins increase the hydrodynamic size of the LNPs by 7-14 nm for ApoE, VTN, and A2M as expected based on previous corona studies.⁸⁻¹⁰ Although the 59 nm and 39 nm increase for CR and the protein mixture is larger, this size change is similar to increases identified for other lipid-based particles incubated with corona proteins.¹¹⁻¹³

Reviewer #3 (Remarks to the Author):

This manuscript investigates the impact of the protein corona on lipid nanoparticle (LNP) uptake and mRNA transfection efficiency. The authors synthesize LNPs, characterize their corona composition, and assess how the most abundant proteins influence transfection.

While this is an interesting study, a major concern is that the findings are largely limited to HepG2 cells, with no validation in more physiologically relevant models such as primary human cells or in vivo systems. Without demonstrating that these effects are generalizable across different biological environments, the broader implications of the study remain uncertain.

While we appreciate the reviewer's viewpoint and agree that understanding how the protein corona influences LNP function in other cell lines is important, extending the characterization to multiple cell lines would require several months, if not years, of additional experimentation. The focus of our paper is to 1) provide a robust experimental protocol to identify hard corona proteins, and 2) to study the effects of these highly enriched hard corona proteins on LNP function. Indeed, we use HepG2 cells as a model cell, one that is often used for high throughput screening of LNP formulations to optimize LNP function. As such, it is a relevant cell model to study the fundamental mechanisms by which hard corona proteins modify the efficacy of LNPs for mRNA delivery and subsequent transfection efficiency.¹⁶⁻¹⁸ Our study highlights that one key output measure that is understood as highly important for LNP performance - cellular internalization - does not correlate and sometimes *inversely* correlates with mRNA expression likely due to the role of hard corona proteins in increasing LNP lysosomal degradation.

We believe these insights to be important even if each individual protein does not translate to the exact outcome in different cell types. In fact, we would expect that to be the case, as perhaps corona proteins also contribute to why different LNP formulations biodistribute differently. Nonetheless, our findings here contribute broadly to the field by encouraging a greater focus on how corona proteins can contribute - positively or negatively - to LNP function. An exciting future direction and separate paper will be a study of how corona proteins can affect biodistribution and cell-type specificity. As we mention above, it addresses a different important question than the focus of the current paper.

Additionally, the observation that protein corona formation impairs mRNA transfection is well established. Serum proteins are known to interfere with transfection, which is why serum-free conditions are commonly used. While the study reinforces this phenomenon, it does not provide novel mechanistic insights or propose innovative strategies to address these limitations.

We thank the reviewer for their comments about serum proteins. In our manuscript, we provide several novel insights including techniques for label-free characterization of the LNP protein corona (Figures 1-2) and mechanistic evidence for proteins driving a mismatch between LNP cellular internalization and mRNA expression (Figures 4-5). That our paper provides concurrent quantification of *increased* LNP internalization and also *decreased* mRNA expression for the same hard corona protein constituent is novel, as is the identification of hard corona proteins on the LNP surface in Figure 3. Our study provides numerous new insights not only in the role of the hard protein corona on LNP efficiency, but also on the mechanism by which corona proteins compromise LNP function.

Specifically, our study presents several new methods and findings: first, we demonstrate that it is possible to use label-free ultracentrifugation and account for the presence of endogenous nanoparticles to selectively identify enriched corona proteins. We achieve this by limiting the presence of endogenous nanoparticles and normalization in the downstream analysis as shown in Figures 2-3. Current techniques

for corona characterization do not account for these endogenous nanoparticles or require modification to the LNP formulation as discussed in lines 132-142 in our manuscript. Next, in Figures 4-5, we show that enriched corona proteins we identified with our workflow increase LNP uptake into the cell, but increased lysosomal trafficking in the presence of these proteins leads to an overall decrease in mRNA expression. This result provides mechanistic evidence that proteins are involved in lysosomal trafficking, a bottleneck in LNP efficiency. While prior work and current standards for LNP transfection favor serum-free conditions as mentioned by our reviewer, our study explains the mechanistic underpinnings of this standard practice.

We discuss these mechanistic findings in more detail in lines 560-579, “The observed mRNA expression and uptake patterns for LNPs with pre-formed ApoE and mixed protein coronas also provide evidence that proteins influence LNP functionality beyond uptake. LNPs pre-incubated with ApoE had no significant increase in mRNA expression but had a five-fold or four-fold increase in cellular uptake in comparison to LNPs without pre-formed coronas according to microscopy and flow cytometry, respectively. Similarly, LNPs pre-incubated with mixed proteins had no significant increase in mRNA expression but had a four-fold increase in cellular uptake in comparison to LNPs without pre-formed coronas as demonstrated by both microscopy and flow cytometry. Co-localization analysis of ApoE-LNPs with lysosomes revealed a 2.9-fold increase in lysosome co-localization for LNPs with pre-formed ApoE coronas relative to LNPs alone, while LNPs with mixed protein corona had a four-fold increase in lysosomal co-localization. In both cases, we find a similar increase in both cell uptake and lysosome co-localization for LNPs with pre-formed ApoE and mixed protein coronas. Specifically, the four-fold increase in cell uptake and the 2.9-fold or four-fold increase in lysosomal co-localization suggests that, as more LNPs enter the cell, more LNPs are also trafficked to the lysosome for degradation. Lysosomal degradation of these LNPs likely accounts for the similar levels of mRNA expression between the ApoE- or mixed corona LNPs and the LNPs without a pre-formed corona. These results suggest that although the presence of an ApoE corona is beneficial for cell uptake into hepatocytes, the ApoE corona may also be inhibiting endosomal escape for this LNP. As these proteins enter the acidic environment of the early endosome, the net negative charge of ApoE shifts to a net positive charge with an isoelectric point of approximately 5.65, potentially influencing ApoE-lipid interactions and affecting endosomal escape.^{68,69}”

To better highlight these aspects of our study, we have added more discussion in lines 581-593 to specify how these findings can be used in future therapeutic design strategies, “Our results provide evidence that enriched LNP corona proteins influence mRNA cargo delivery beyond cellular internalization, suggesting a more complex mechanism of LNP endosomal escape.⁷⁰ Future therapeutic development requires further study of how protein-LNP interactions govern this key endosomal escape bottleneck in limiting LNP efficiency, yet our study highlights the contribution of the protein corona in hindering endosomal escape. Additionally, LNP design would benefit from considering how LNP corona enriched proteins and/or their peptides that enhance cellular internalization may be leveraged to simultaneously mitigate trafficking to the lysosome. We can then design LNPs with favorable biomolecular interactions to optimize LNP function as demonstrated recently by the development of a prototype apolipoprotein-based nanoparticle that leverages native lipoprotein trafficking as a delivery strategy.⁷¹ Alternatively, strategies to prevent the interaction of specific proteins that activate inflammation could be considered. Collectively, these findings highlight that understanding the protein corona is important for the rational design of LNPs to overcome delivery bottlenecks at the points of cell entry and endosomal escape.”

Further, the role of vitronectin and other proteins in endosomal escape and transfection efficiency remains insufficiently explored. Although vitronectin enhances LNP uptake, it paradoxically reduces mRNA transfection efficiency. However, no significant differences in lysosomal colocalization were observed, casting doubt on the proposed mechanism. Additional experiments are needed to clarify this finding.

We thank the reviewer and agree that this area of the paper requires further experimental exploration and validation. We found that the additional flow cytometry experiments in Figure 5 provided further evidence to support our proposed explanation. We believe that these new experiments clarify our findings, which we discuss further in lines 541-558, “Counterintuitively, we found that although LNPs incubated with VTN or CR displayed decreased levels of mRNA expression, they did not have decreased levels of cell uptake (Fig. 6a). In fact, VTN-LNPs showed increased cell uptake relative to LNPs not pre-incubated with protein, while LNPs incubated with CR had no significant difference in uptake relative to LNPs without a pre-formed corona according to our microscopy results. We hypothesized that this increase in cellular uptake for VTN-LNPs may be due to their association with the membrane rather than internalization into the cell cytoplasm, as our localization analysis supported the conclusion that VTN-LNPs generally adhere more to the outside of the cell relative to protein-free LNPs. Specifically, VTN-LNPs, when compared to LNPs incubated without protein, show 8.5% more signal localized to the outer region of the cell versus inside the cell, suggesting that the association with the outside of the cell may prevent effective cargo delivery, leading to decreased mRNA expression. To further investigate VTN-LNP uptake, we applied flow cytometry to quantify cell uptake of LNPs with pre-formed VTN coronas after a wash step, which removes LNPs bound to the outside of the cell. We found that although VTN-LNPs still show slightly more cell uptake than LNPs without a pre-formed corona, the difference between the two conditions is smaller and not statistically significant. Together, these results suggest that LNPs pre-incubated with VTN may exhibit lower mRNA expression partially due to LNP adhesion to the outer cell membrane. Furthermore, VTN corona proteins may have an additional impact on LNP endosomal escape, as we observe a small but not statistically significant increase in lysosomal co-localization for LNPs with pre-formed VTN coronas (Fig. 5g).”

Major Concerns:

- The study lacks crucial details on the quantities of LNPs and plasma used for corona formation, making it difficult to assess how accurately the conditions mimic *in vivo* environments. This omission weakens the study's biological relevance.

We thank the reviewer for their comments. As specified in the methods section in line 685, “LNPs synthesized at a concentration of 0.05 mg/mL mRNA with Cre recombinase mRNA were incubated with an equal volume (400 μ L) of pooled human blood plasma at 37 °C, the physiologically relevant temperature, for 1 hour, which has previously been determined as sufficient time for corona formation to occur.” Additionally, quantities of the LNPs were listed under LNP synthesis, in line 668, “Lipid nanoparticles for protein corona isolation and *in vitro* studies were formulated at final mRNA concentrations of 0.05 and 0.01 mg/mL mRNA, respectively.” These 1:1 volume plasma to LNP incubation conditions were selected based on well-established incubation conditions for protein corona formation and specifically LNP protein corona formation.^{3,14,15} Furthermore, the selected LNP concentrations mirror the LNP concentrations used for *in vivo* injections of this model formulation.¹⁶ If we have omitted additional details that require inclusion for a better understanding of our experimental protocols, we welcome additional feedback.

- The authors mention that most of the LNPs were localized in fraction 2-6 of the iodixanol gradient, whereas most of the endogenous particles were present in fractions 5-10. But this analysis was based solely on the amount of cholesterol, as it has been considered as the key lipoprotein constituent. This assumption needs to be backed by more quantification.

In our manuscript, we show that cholesterol, a constituent of lipoproteins, localizes to fractions 5-10, indicating that across all types of lipoproteins localization is limited in the region we collect samples. Cholesterol is only present in human blood plasma within lipoproteins, and therefore we are confident that

our approach minimizes contamination from lipoproteins. We nonetheless recognize the importance of orthogonally validating our claims that our method enables recovery of corona proteins predominantly from LNPs. Accordingly, we have performed additional control experiments to strengthen our analysis. We performed an enzyme-linked immunosorbent assay (ELISA) to quantify human apolipoprotein A-I within the density gradient as an example apolipoprotein. In this antibody-based approach, we confirm that the majority of apolipoprotein AI, 99.5%, is present in fractions 12-24, and that apolipoprotein AI is limited in fractions 2-6. We added the figure to the Supplemental Information and updated the main text to reflect this additional data in lines 200-203, "To further validate the localization of lipoproteins, we quantified the presence of the most abundant apolipoprotein in human plasma, apolipoprotein AI (ApoA-I). We confirmed that 99.5% of ApoA-I localizes to fractions 12-24 through an enzyme-linked immunosorbent assay (ELISA; Supplementary Fig. 5).⁵⁴"

Supplementary Figure 5. Apolipoprotein AI localization within the density gradient. Average human apolipoprotein AI enzyme-linked immunosorbent assay quantification of plasma alone gradient fractions collected after DGC isolation workflow shows that 99.5% of apolipoprotein AI proteins are present among fractions 12-24. We estimated the percentage of lipoprotein A within the peak by calculating the area under the measured fluorescence curve for fractions 12-24 relative to the total area under the measured fluorescence curve using the trapezoidal rule (trapz function from scipy.integrate). N = 2 biological replicates. Error bars all denote standard deviation.

Additionally, as mentioned in lines 173-174, this protocol is based on methods used to separate subpopulations of exosomes. As such, the presence of exosomes within these density gradient conditions has been characterized in Temoche-Diaz et. al's work,¹ which shows exosome localization in their Figure 1b through a immunoblot of key exosome markers. In their work, 0.4 mL fractions are collected, while we collect 0.5 mL. They found no exosomes present in fractions 1-6 (equivalent to fractions 1-5 in our study)

and the presence of low density exosomes in fractions 9-11 (equivalent to fractions 7-9 in our study). To clarify the position of exosomes within the density gradient, we have modified the manuscript text in lines 203-205 to “Previous work has already determined that exosomes, another type of endogenous particle, are not present in the first 5 fractions of the density gradient.”⁵⁴

Figure 1b from Temoche-Diaz, M. M.; Shurtleff, M. J.; Nottingham, R. M.; Yao, J.; Fadadu, R. P.; Lambowitz, A. M.; Schekman, R. Distinct Mechanisms of microRNA Sorting into Cancer Cell-Derived Extracellular Vesicle Subtypes. *Elife* 2019, 8, e47544.

- The authors mention the current limitations for protein corona quantification on LNPs. Specifically, it focuses on two key points : firstly, it highlights the importance of excluding the nano-sized particles which are intrinsically present in the biological fluids, and secondly it also talks about the low density of the soft nanoparticles, which causes difficulty to isolate protein-LNP complex using centrifugation. The authors back their claim in Supplementary Fig - 1A, where the ultracentrifugation was performed for 2 hours at 160,000 relative centrifugal force. We acknowledge and appreciate the need to quantify protein corona in a robust manner, however we recognize that the unbound proteins can also be eluted by ultracentrifugation for lesser time (e.g. 30 minutes) at a much lesser rcf as it has been established and verified previously in other studies. This questions the novelty of the work and further raises concerns about the robustness of the study.

We thank the reviewer for their comments, and we counter that current ultracentrifugation methods remain a controversial approach to characterizing the LNP protein corona. This is highlighted by recently published methods papers that explore the limitations of centrifugal approaches for LNP corona isolation.^{3,19,20} Simple ultracentrifugation sediments all particles to the bottom of the centrifugation tube, which makes it impossible to distinguish between proteins interacting with LNPs and those interacting with lipoprotein particles. This artifact has limited our understanding of which proteins are highly enriched on the LNP surface, and specific lipoproteins, such as Apolipoprotein E, are involved in organ targeting for LNP formulations, further increasing the significance of understanding which lipoproteins are enriched in the LNP protein corona. Our approach also entails novelty in our analysis: only considering corona proteins that are repeatedly enriched across numerous technical *and* experimental replicates increases the chance that they are indeed LNP protein corona constituents. Through differential analysis of our LNP gradient sample, we are able to limit the presence of endogenous particles during isolation of LNPs in the

density gradient as demonstrated with cholesterol and lipoprotein A quantification. We would also like to highlight that as illustrated in Supplementary Figure 6, we select for fractions maximizing concentrations of these LNPs and we compare to a plasma alone control. As such, the combined novelty of our approach includes advances both in the ultracentrifugation and in the comparative analysis, uniquely allowing identification of select apolipoproteins but not the most abundant apolipoproteins. This enables us to account for this contamination as discussed in lines 258-261, “Despite ApoA-I and apolipoprotein A-II (ApoA-II) being the two most abundant apolipoproteins in blood plasma,⁵⁸ we do not identify ApoA-I or ApoA-II as enriched in the protein corona, suggesting we are selectively identifying apolipoproteins that interact with LNPs.” In turn, we can reliably identify *only* corona proteins that form the hard corona and selectively study their effects on LNP function.

- The in vitro studies focused on the four most abundant proteins identified in the protein corona. While understanding the individual contributions of these proteins is valuable, a comprehensive assessment of the corona's impact on LNP functionality requires considering all the hard corona proteins.

We thank the reviewer for their interest in a comprehensive assessment of the corona's impact on LNP functionality. We are unsure if the reviewer is i) requesting additional experiments to further confirm with increased certainty that our proteomic analysis is truly allowing us to characterize all the hard corona proteins, or ii) asking us to consider the interactions of all the hard corona proteins as a mixture to provide a more comprehensive assessment of the corona's impact on the LNP functionality. For the former, we performed a post-hoc power analysis using Gpower software to demonstrate that we have sufficient power for our analysis of LNP internalization as an example. We had 3 biological replicates per 2 conditions for this analysis for a total sample size of 6. We assumed the Alpha error prob to be 0.05 and that we were only considering 2 groups because in our analysis we only compare between the control and the individual protein group. We find that in both these cases we have sufficient power based on our analysis (<0.80). We find that we have a relatively large effect size for these observations, and with the same number of samples used for the identification of hard corona proteins, we can be confident that we have analyzed a sufficiently large sample to have identified the hard corona proteins most tightly associated with the LNPs.

Example data

	LNP	LNP + VTN	LNP + ApoE
Rep 1	25679	128055	252939
Rep 2	35774	173393	147160
Rep 3	36564	116874	142957
Average	32672	139441	181019
Standard Deviation	6069	29930	62320
Pooled standard deviation		15269.75	31307.56
Effect size f		1.56	1.8

Power

0.812

0.902

For the latter, we have now studied how the protein mixture influences the internalization of the LNP into the cell and lysosomal trafficking. We have updated the results and discussion sections to include these protein mixture corona experiments, which further support our findings that specific coronas increase cellular internalization without increasing mRNA expression. We discuss these results in lines 560-579, “The observed mRNA expression and uptake patterns for LNPs with pre-formed ApoE and mixed protein coronas also provide evidence that proteins influence LNP functionality beyond uptake. LNPs pre-incubated with ApoE had no significant increase in mRNA expression but had a five-fold or four-fold increase in cellular uptake in comparison to LNPs without pre-formed coronas according to microscopy and flow cytometry, respectively. Similarly, LNPs pre-incubated with mixed proteins had no significant increase in mRNA expression but had a four-fold increase in cellular uptake in comparison to LNPs without pre-formed coronas as demonstrated by both microscopy and flow cytometry. Co-localization analysis of ApoE-LNPs with lysosomes revealed a 2.9-fold increase in lysosome co-localization for LNPs with pre-formed ApoE coronas relative to LNPs alone, while LNPs with mixed protein corona had a four-fold increase in lysosomal co-localization. In both cases, we find a similar increase in both cell uptake and lysosome co-localization for LNPs with pre-formed ApoE and mixed protein coronas. Specifically, the four-fold increase in cell uptake and the 2.9-fold or four-fold increase in lysosomal co-localization suggests that, as more LNPs enter the cell, more LNPs are also trafficked to the lysosome for degradation. Lysosomal degradation of these LNPs likely accounts for the similar levels of mRNA expression between the ApoE- or mixed corona LNPs and the LNPs without a pre-formed corona. These results suggest that although the presence of an ApoE corona is beneficial for cell uptake into hepatocytes, the ApoE corona may also be inhibiting endosomal escape for this LNP. As these proteins enter the acidic environment of the early endosome, the net negative charge of ApoE shifts to a net positive charge with an isoelectric point of approximately 5.65, potentially influencing ApoE-lipid interactions and affecting endosomal escape.^{68,69} “

- Internalization studies are based solely on imaging. Incorporating flow cytometry or other quantification methods would provide a more rigorous analysis.

We thank the reviewer for their comments, and we agree that additional approaches to quantification would be more rigorous. We applied flow cytometry to assess whether the trends we observe during confocal microscopy hold. Indeed, we find that the trends we observe during confocal microscopy image analysis are consistent with the trends we observe from flow cytometry. We have modified Figure 5 to include this supporting data in lines 465-482, “To further validate differences in internalization for LNPs with pre-formed coronas observed during confocal microscopy, we used flow cytometry to measure cellular internalization of fluorescently labeled LNPs (Fig. 5d-e, Supplementary Fig. 11a-b). LNPs with and without a pre-formed corona were incubated with HepG2 cells for 1 hour at 37 °C before cells were washed to remove LNPs on the outer surface. We found that uptake trends quantified with flow cytometry are consistent with the confocal microscopy data for both the percentage of Cy5 positive cells (Fig. 5d) and the difference in mean Cy5 fluorescence intensity between cells incubated with LNPs with a pre-formed protein corona compared to LNPs without a preformed corona (Fig. 5e). Specifically, we observe that cells exposed to LNPs pre-incubated with the protein mixture had four-fold higher levels of Cy5 mean fluorescence intensity than those exposed to LNPs without a preformed corona. Cells exposed to LNPs with an ApoE or VTN corona had a four-fold and 1.9-fold higher mean fluorescence intensity, respectively, though this difference was not statistically significant. In contrast, cells exposed to LNPs pre-incubated with A2M or CR had similarly low levels of mean fluorescence intensity as cells exposed to

LNPs alone. These results provide an orthogonal method of validating our microscopy data, further supporting aforementioned LNP internalization trends showing that certain protein coronas increase cellular uptake of LNPs. This counterintuitive result that certain single-component pre-formed protein coronas and the select corona mixtures increase cell uptake while decreasing transfection efficiency suggests that corona proteins may affect the efficiency of LNP endosomal escape."

- The findings indicate greater internalization of VTN+LNP compared to LNP without a corona. However, an unexpected discrepancy was observed with mRNA transfection, showing lower transfection efficiency for VTN+LNP than for LNP without a corona. This suggests that while vitronectin enhances cellular uptake, it may hinder endosomal escape, potentially leading to reduced mRNA transfection. However, no significant differences were observed in lysosomal colocalization between VTN+LNP and LNP without a corona, raising questions about this conclusion. Further studies are necessary to clarify how vitronectin influences endosomal escape and transfection efficiency.

We thank the reviewer for these insightful comments, and we appreciate that more in-depth analysis of the mechanism is required. As stated in lines 459-461, we did hypothesize that this mismatch is due to adhesion to the outer surface. Our erosion analysis in combination with our recently added flow cytometry experiments suggest that these observed differences are due to vitronectin's adhesion to the outer membrane surface. We believe that these new experiments clarify these findings, which we discuss further in lines 541-558, "Counterintuitively, we found that although LNPs incubated with VTN or CR displayed decreased levels of mRNA expression, they did not have decreased levels of cell uptake (Fig. 6a). In fact, VTN-LNPs showed increased cell uptake relative to LNPs not pre-incubated with protein, while LNPs incubated with CR had no significant difference in uptake relative to LNPs without a pre-formed corona according to our microscopy results. We hypothesized that this increase in cellular uptake for VTN-LNPs may be due to their association with the membrane rather than internalization into the cell cytoplasm, as our localization analysis supported the conclusion that VTN-LNPs generally adhere more to the outside of the cell relative to protein-free LNPs. Specifically, VTN-LNPs, when compared to LNPs incubated without protein, show 8.5% more signal localized to the outer region of the cell versus inside the cell, suggesting that the association with the outside of the cell may prevent effective cargo delivery, leading to decreased mRNA expression. To further investigate VTN-LNP uptake, we applied flow cytometry to quantify cell uptake of LNPs with pre-formed VTN coronas after a wash step, which removes LNPs bound to the outside of the cell. We found that although VTN-LNPs still show slightly more cell uptake than LNPs without a pre-formed corona, the difference between the two conditions is smaller and not statistically significant. Together, these results suggest that LNPs pre-incubated with VTN may exhibit lower mRNA expression partially due to LNP adhesion to the outer cell membrane. Furthermore, VTN corona proteins may have an additional impact on LNP endosomal escape, as we observe a small but not statistically significant increase in lysosomal co-localization for LNPs with pre-formed VTN coronas (Fig. 5g)."

References:

- (1) Temoche-Diaz, M. M.; Shurtleff, M. J.; Nottingham, R. M.; Yao, J.; Fadadu, R. P.; Lambowitz, A. M.; Schekman, R. Distinct Mechanisms of microRNA Sorting into Cancer Cell-Derived Extracellular Vesicle Subtypes. *Elife* **2019**, *8*, e47544. <https://doi.org/10.7554/eLife.47544>.
- (2) Pattipeiluhu, R.; Crielgaard, S.; Klein-Schiphorst, I.; Florea, B. I.; Kros, A.; Campbell, F. Unbiased Identification of the Liposome Protein Corona Using Photoaffinity-Based Chemoproteomics. *ACS Cent. Sci.* **2020**, *6* (4), 535–545. <https://doi.org/10.1021/acscentsci.9b01222>.
- (3) Francia, V.; Zhang, Y.; Cheng, M. H. Y.; Schiffelers, R. M.; Witzigmann, D.; Cullis, P. R. A Magnetic Separation Method for Isolating and Characterizing the Biomolecular Corona of Lipid Nanoparticles. *Proceedings of the National Academy of Sciences* **2024**, *121* (11), e2307803120. <https://doi.org/10.1073/pnas.2307803120>.

- (4) Ma, Y.; VanKeulen-Miller, R.; Fenton, O. S. mRNA Lipid Nanoparticle Formulation, Characterization and Evaluation. *Nat Protoc* **2025**, 1–34. <https://doi.org/10.1038/s41596-024-01134-4>.
- (5) Swingle, K. L.; Hamilton, A. G.; Safford, H. C.; Geisler, H. C.; Thatte, A. S.; Palanki, R.; Murray, A. M.; Han, E. L.; Mukalel, A. J.; Han, X.; Joseph, R. A.; Ghalsasi, A. A.; Alameh, M.-G.; Weissman, D.; Mitchell, M. J. Placenta-Tropic VEGF mRNA Lipid Nanoparticles Ameliorate Murine Pre-Eclampsia. *Nature* **2025**, 637 (8045), 412–421. <https://doi.org/10.1038/s41586-024-08291-2>.
- (6) Tabatabaeian Nimavard, R.; Sadeghi, S. A.; Mahmoudi, M.; Zhu, G.; Sun, L. Top-Down Proteomic Profiling of Protein Corona by High-Throughput Capillary Isoelectric Focusing-Mass Spectrometry. *J. Am. Soc. Mass Spectrom.* **2025**, 36 (4), 778–786. <https://doi.org/10.1021/jasms.4c00463>.
- (7) Winzen, S.; Schoettler, S.; Baier, G.; Rosenauer, C.; Mailaender, V.; Landfester, K.; Mohr, K. Complementary Analysis of the Hard and Soft Protein Corona: Sample Preparation Critically Effects Corona Composition. *Nanoscale* **2015**, 7 (7), 2992–3001. <https://doi.org/10.1039/C4NR05982D>.
- (8) Casals, E.; Pfaller, T.; Duschl, A.; Oostingh, G. J.; Puentes, V. Time Evolution of the Nanoparticle Protein Corona. *ACS Nano* **2010**, 4 (7), 3623–3632. <https://doi.org/10.1021/nn901372t>.
- (9) Palchetti, S.; Colapicchioni, V.; Digiacoio, L.; Caracciolo, G.; Pozzi, D.; Capriotti, A. L.; La Barbera, G.; Laganà, A. The Protein Corona of Circulating PEGylated Liposomes. *Biochimica et Biophysica Acta (BBA) - Biomembranes* **2016**, 1858 (2), 189–196. <https://doi.org/10.1016/j.bbmem.2015.11.012>.
- (10) Röcker, C.; Pötzl, M.; Zhang, F.; Parak, W. J.; Nienhaus, G. U. A Quantitative Fluorescence Study of Protein Monolayer Formation on Colloidal Nanoparticles. *Nature Nanotech* **2009**, 4 (9), 577–580. <https://doi.org/10.1038/nnano.2009.195>.
- (11) Caracciolo, G.; Pozzi, D.; Capriotti, A. L.; Cavaliere, C.; Piovesana, S.; Barbera, G. L.; Amici, A.; Laganà, A. The Liposome–Protein Corona in Mice and Humans and Its Implications for in Vivo Delivery. *J. Mater. Chem. B* **2014**, 2 (42), 7419–7428. <https://doi.org/10.1039/C4TB01316F>.
- (12) Caracciolo, G.; Pozzi, D.; L. Capriotti, A.; Cavaliere, C.; Piovesana, S.; Amenitsch, H.; Laganà, A. Lipid Composition: A “Key Factor” for the Rational Manipulation of the Liposome–Protein Corona by Liposome Design. *RSC Advances* **2015**, 5 (8), 5967–5975. <https://doi.org/10.1039/C4RA13335H>.
- (13) Corbo, C.; Molinaro, R.; Taraballi, F.; Toledano Furman, N. E.; Sherman, M. B.; Parodi, A.; Salvatore, F.; Tasciotti, E. Effects of the Protein Corona on Liposome–Liposome and Liposome–Cell Interactions. *Int J Nanomedicine* **2016**, 11, 3049–3063. <https://doi.org/10.2147/IJN.S109059>.
- (14) Dilliard, S. A.; Cheng, Q.; Siegwart, D. J. On the Mechanism of Tissue-Specific mRNA Delivery by Selective Organ Targeting Nanoparticles. *Proc Natl Acad Sci USA* **2021**, 118 (52), e2109256118. <https://doi.org/10.1073/pnas.2109256118>.
- (15) Barrán-Berdón, A. L.; Pozzi, D.; Caracciolo, G.; Capriotti, A. L.; Caruso, G.; Cavaliere, C.; Riccioli, A.; Palchetti, S.; Laganà, A. Time Evolution of Nanoparticle–Protein Corona in Human Plasma: Relevance for Targeted Drug Delivery. *Langmuir* **2013**, 29 (21), 6485–6494. <https://doi.org/10.1021/la401192x>.
- (16) LoPresti, S. T.; Arral, M. L.; Chaudhary, N.; Whitehead, K. A. The Replacement of Helper Lipids with Charged Alternatives in Lipid Nanoparticles Facilitates Targeted mRNA Delivery to the Spleen and Lungs. *Journal of Controlled Release* **2022**, 345, 819–831. <https://doi.org/10.1016/j.jconrel.2022.03.046>.
- (17) Zhu, Y.; Shen, R.; Vuong, I.; Reynolds, R. A.; Shears, M. J.; Yao, Z.-C.; Hu, Y.; Cho, W. J.; Kong, J.; Reddy, S. K.; Murphy, S. C.; Mao, H.-Q. Multi-Step Screening of DNA/Lipid Nanoparticles and Co-Delivery with siRNA to Enhance and Prolong Gene Expression. *Nat Commun* **2022**, 13 (1), 4282. <https://doi.org/10.1038/s41467-022-31993-y>.
- (18) Kim, M.; Jeong, M.; Hur, S.; Cho, Y.; Park, J.; Jung, H.; Seo, Y.; Woo, H. A.; Nam, K. T.; Lee, K.; Lee, H. Engineered Ionizable Lipid Nanoparticles for Targeted Delivery of RNA Therapeutics into Different Types of Cells in the Liver. *Science Advances* **2021**, 7 (9), eabf4398. <https://doi.org/10.1126/sciadv.abf4398>.
- (19) Francia, V.; Schiffelers, R. M.; Cullis, P. R.; Witzigmann, D. The Biomolecular Corona of Lipid Nanoparticles for Gene Therapy. *Bioconjugate Chem.* **2020**, 31 (9), 2046–2059. <https://doi.org/10.1021/acs.bioconjchem.0c00366>.
- (20) Simonsen, J. B.; Münter, R. Pay Attention to Biological Nanoparticles When Studying the Protein Corona on Nanomedicines. *Angewandte Chemie International Edition* **2020**, 59 (31), 12584–12588. <https://doi.org/10.1002/anie.202004611>.